# A barbed end interference mechanism reveals how capping protein promotes nucleation in branched actin networks

Johanna Funk[1,6], Felipe Merino[2,3,6], Matthias Schaks[4,5], Klemens Rottner 🔘 [4,5], Stefan Raunser 🔘 [2✉] & Peter Bieling 🔘 [1✉]

Heterodimeric capping protein (CP/CapZ) is an essential factor for the assembly of branched actin networks, which push against cellular membranes to drive a large variety of cellular processes. Aside from terminating filament growth, CP potentiates the nucleation of actin filaments by the Arp2/3 complex in branched actin networks through an unclear mechanism. Here, we combine structural biology with in vitro reconstitution to demonstrate that CP not only terminates filament elongation, but indirectly stimulates the activity of Arp2/3 activating nucleation promoting factors (NPFs) by preventing their association to filament barbed ends. Key to this function is one of CP's C-terminal "tentacle" extensions, which sterically masks the main interaction site of the terminal actin protomer. Deletion of the β tentacle only modestly impairs capping. However, in the context of a growing branched actin network, its removal potently inhibits nucleation promoting factors by tethering them to capped filament ends. End tethering of NPFs prevents their loading with actin monomers required for activation of the Arp2/3 complex and thus strongly inhibits branched network assembly both in cells and reconstituted motility assays. Our results mechanistically explain how CP couples two opposed processes—capping and nucleation—in branched actin network assembly.

[1] Department of Systemic Cell Biology, Max Planck Institute of Molecular Physiology, Dortmund, Germany. [2] Department of Structural Biochemistry, Max Planck Institute of Molecular Physiology, Dortmund, Germany. [3] Department of Protein Evolution, Max Planck Institute for Developmental Biology, Tübingen, Germany. [4] Division of Molecular Cell Biology, Zoological Institute, Technische Universität Braunschweig, Braunschweig, Germany. [5] Department of Cell Biology, Helmholtz Centre for Infection Research, Braunschweig, Germany. [6]These authors contributed equally: Johanna Funk, Felipe Merino. ✉email: StefanRaunser@mpi-dortmundmpgde; PeterBieling@mpi-dortmundmpgde

Heterodimeric capping protein (CP/CapZ) is an evolutionarily ancient actin regulator found in nearly all eukaryotic organisms and cell types[1–4] that controls the fate and interactions of actin filament ends in a variety of cellular contexts. Comprised of two closely related α and β subunits, it forms a constitutive heterodimer that tightly binds to the barbed end of actin filaments to terminate their growth. Within sarcomeres, it caps the barbed ends of filaments in the z-disk. In non-muscle cells, CP collaborates with a core set of proteins—profilin, the Arp2/3 complex and a membrane-localized nucleation promoting factor (NPF) of the WASP protein family—to create dense, polarized networks of short and branched filaments[5–7]. These branched actin networks generate pushing forces to move cellular membranes[8–10] in a large number of cellular processes ranging from cell motility[11], endocytosis[12], phagocytosis[13], autophagy[14] to cell-cell adhesion[15]. CP's central role in branched network assembly is commonly thought to simply result from its eponymous biochemical activity— to cap actin filament barbed ends and inhibit the addition of further subunits[2,16–18]. However, in vitro reconstitutions of branched network assembly suggest a more complex, multi-functional role[5].

Capping protein is known to contribute to branched actin network assembly in several ways. First, it prevents the elongation of filaments beyond a critical length at which they would easily buckle under load[6,19,20]. Second, it rapidly stops the non-productive elongation of filaments distant from the cell membrane. This generates a requirement for continuous nucleation of new filament branches via the Arp2/3 complex, which is in turn activated by a membrane-bound NPF. Confinement of Arp2/3 complex activation to the membrane is therefore responsible for polarized network growth[5,21]. In addition to these established functions directly related to the termination of filament growth, CP has been shown to stimulate nucleation of new filaments by the Arp2/3 complex. Perturbing CP activity either in vivo or in reconstituted actin motility results in comparable effects that are consistent with it acting as a positive regulator of Arp2/3-dependent branching nucleation[5,7,22–24]. Yet, the responsible biochemical mechanism has remained enigmatic.

Branching nucleation itself is a multi-step process that requires the transient formation of a higher-order complex between the Arp2/3 complex bound to the side of an actin mother filament and a NPF that is loaded with an actin monomer bound to its WH2 domain[25]. It is presently unclear, which steps in this process control the rate of filament nucleation within growing actin networks. The activity of CP seems to be a key factor[5], but we currently do not understand how CP might either directly or indirectly influence the Arp2/3 complex, its upstream NPF or any intermediates formed between the two during nucleation. Prior structural and biochemical work on CP provides little insight into this crucial question[2]. One potential hint comes from the observation that each CP subunit possesses a C-terminal helical extension that protrudes from the body of the heterodimer known as tentacle[26]. Both tentacles share sequence similarity with the actin-binding WH2 domain of NPFs. Based on a low-resolution structure of capped skeletal muscle actin filaments[27], it has been proposed that most of the interaction occurs between the body of CP, which, together with its α tentacle, contacts the penultimate actin protomer. In addition, the β tentacle constitutes a potential secondary actin-binding region that has been speculated to bind to the last actin subunit[2,27–29]. While this is indeed the case in the homologous complex found within the dynactin complex, in which CP caps a short Arp1 filament[30], the structural details of capped actin filaments are still unknown. As a consequence, we understand very little how CP might influence Arp2/3 complex activity.

One conceptual model of why CP stimulates nucleation in branched networks is offered by the so-called monomer gating hypothesis[5,24]. Key to monomer gating is a presumed kinetic competition between growing filament ends and the Arp2/3 complex for a limiting flux of actin monomers already in complex with membrane-bound NPFs. As polymerizing filament ends become more numerous through branching, they should consume larger amounts of NPF-bound monomers, which might reduce their partitioning into nucleation via the Arp2/3 complex. While this model offers a conceptual explanation for the nucleation stimulating function of CP, it lacks direct experimental evidence. Indeed, alternative mechanisms based on the local depletion of soluble actin monomers have also been proposed[31].

We are currently lacking a comprehensive mechanism that explains and unifies the multiple roles of CP based on its interactions with the other core components of branched actin networks. Here, we combine electron cryo-microscopy (Cryo-EM) with in vitro reconstitution and cell biology to establish that CP not only prevents the elongation of actin filaments as commonly thought. Instead, we show that it selectively masks the actin filament end to control interactions with other key proteins via a conserved actin-binding motif –the β tentacle. Its removal does not prevent capping, but dramatically inhibits the assembly of Arp2/3-generated actin networks in vitro. In line with this, we find that the CP β tentacle is selectively required for efficient lamellipodial protrusion of mammalian cells and localization of CP to the leading edge. We uncover the responsible molecular mechanism by demonstrating that NPFs activating the Arp2/3 complex are tethered and inactivated by capped filament ends when this binding motif is lost on capping protein. Our results reveal an unanticipated mechanism at the core of the branched actin network engine that couples the creation of polymerizing filaments by Arp2/3 complexes to their elimination through capping protein.

## Results

**The structure of CP-bound actin filament barbed ends**. The ends of actin filaments have emerged as important regulatory sites, to which a diverse set of regulators bind to control filament dynamics. Despite their obvious importance, there are—with a single exception[32]—no high-resolution structures of filament ends available. Because filament elongation is kinetically far more favorable than nucleation, the exceedingly sparse density of ends in common actin filament preparations has prevented single particle analysis until now. To overcome this fundamental limitation, we established a general procedure to generate short actin filaments amenable to single-particle Cryo-EM analysis (Fig. 1A). To this end, we premixed CP with actin in buffers of low ionic strength containing $Ca^{2+}$, which keep it from polymerizing. We then triggered rapid filament nucleation by salt addition. Under these conditions, and in the absence of monomer-binding proteins such as profilin, CP potently nucleates filaments that slowly elongate from the pointed end[33]. We stabilized these capped filament stubs using phalloidin shortly after the initiation of polymerization. To separate stabilized filaments from residual soluble capping protein and actin, we subjected these reactions to size exclusion chromatography (Fig. 1B). SDS PAGE revealed the presence of both CP and actin at roughly 1:20 stoichiometry in the high molecular weight fractions (Fig. 1B, Supplementary Figure 1).

Cryo-EM images of samples from these fractions revealed the presence of densely packed filaments of near-uniform lengths of 80–120 nm (Fig. 1C, Supplementary Figure 1). Using these samples and an iterative classification strategy, we determined a

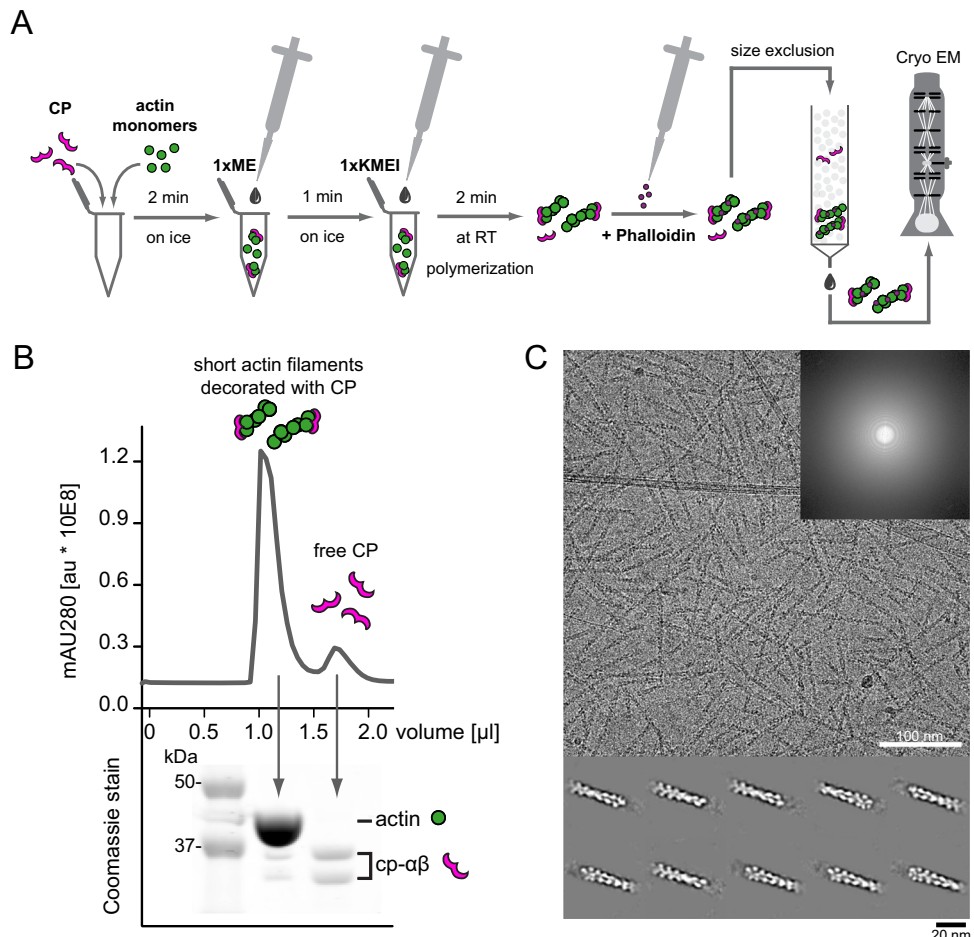

**Fig. 1 Preparation and visualization of CP-bound barbed ends by cryo-EM. A** Workflow of capped filament preparation for cryo-EM. Monomers were mixed with CP followed by $Ca^{2+}$-to-$Mg^{2+}$ exchange and salt addition to trigger polymerization. Polymerization was arrested by phalloidin addition and capped filaments were separated from free CP by size-exclusion chromatography (SEC) and then visualized by cryo-EM. **B** Isolation of short, capped actin-filaments from free CP by SEC from a representative run. The experiment was repeated at least 5 times with similar results. Top: chromatography profile. Bottom: Corresponding Coomassie-stain with bands representing actin or α and β-capping protein as indicated (Supplementary Fig. 1). **C** Representative micrograph of vitrified capped filaments on graphene oxide grids imaged at 1.5 μm defocus. Similar results were obtained in at least five independent replicates. The inset shows the corresponding power spectrum for the image. Bottom: Class averages showing densities for capping protein bound to respective filament barbed ends.

Cryo-EM structure of CP-bound cytoplasmic actin barbed ends to an overall resolution of 3.8 Å (Fig. 2A, Supplementary Fig. 2, Supplementary Table 1). Local resolution ranges from 3.5–7.5 Å, with most actin monomers resolved better than 4 Å and CP to 6–7 Å (Supplementary Fig. 2). We attribute this large variation in map quality to two factors. (i) CP and individual actin monomers are similar in size, making it extremely challenging to differentiate between capped and uncapped filaments during classification. (ii) The directional FSC analysis shows preferential orientation of the particles (Supplementary Fig. 2). Combined with the observation that not all ends are capped (Supplementary Fig. 1), this suggests that ends might become uncapped when CP interacts with the grids in a certain orientation.

As expected, the structure of cytoplasmic actin filaments closely resembles the previously determined structures of skeletal actin[34,35]. We observed clear density for phalloidin in all the expected binding sites[36], with the exception of the cleft between the ultimate and penultimate actin subunits (Supplementary Fig. 2). This is likely due to the absence of the missing third actin protomer required to form the full binding site. In agreement with our previous work[34,36], we find the D-loop in a closed state. Due to the comparably high resolution within the nucleotide-binding site of actin, we could

directly determine the identity of the nucleotide bound to each subunit. Consistent with the early addition of phalloidin during the polymerization reaction, most subunits show clear density for inorganic phosphate (Pi) (Supplementary Fig. 2). Interestingly, the two terminal barbed end subunits lack this density, suggesting that they are in an ADP-bound state. Whether this was due to the direct interaction with CP favoring Pi release or simply the result of our filament preparation procedure, in which these two subunits are the "oldest" in the filament as they form the nucleus for polymerization with CP is unclear.

The primary interaction between CP and the filament occurs at the barbed end face of the penultimate and the inner filament side of the terminal actin subunit (Fig. 2B). The overall interaction between CP and the actin barbed end is very similar to what has been observed in the dynactin complex[30,37] (Supplementary Fig. 3). In spite of the sequence differences between Arp1 and actin, their complexes with CP superimpose almost perfectly (Supplementary Fig. 3), with both tentacles occupying equivalent positions in contacting one end of the Arp1 filament (Supplementary Fig. 3). The core of CP together with the α tentacle block all of the exposed barbed end surface of the penultimate actin protomer (Fig. 2, Supplementary Fig. 3), with the α tentacle

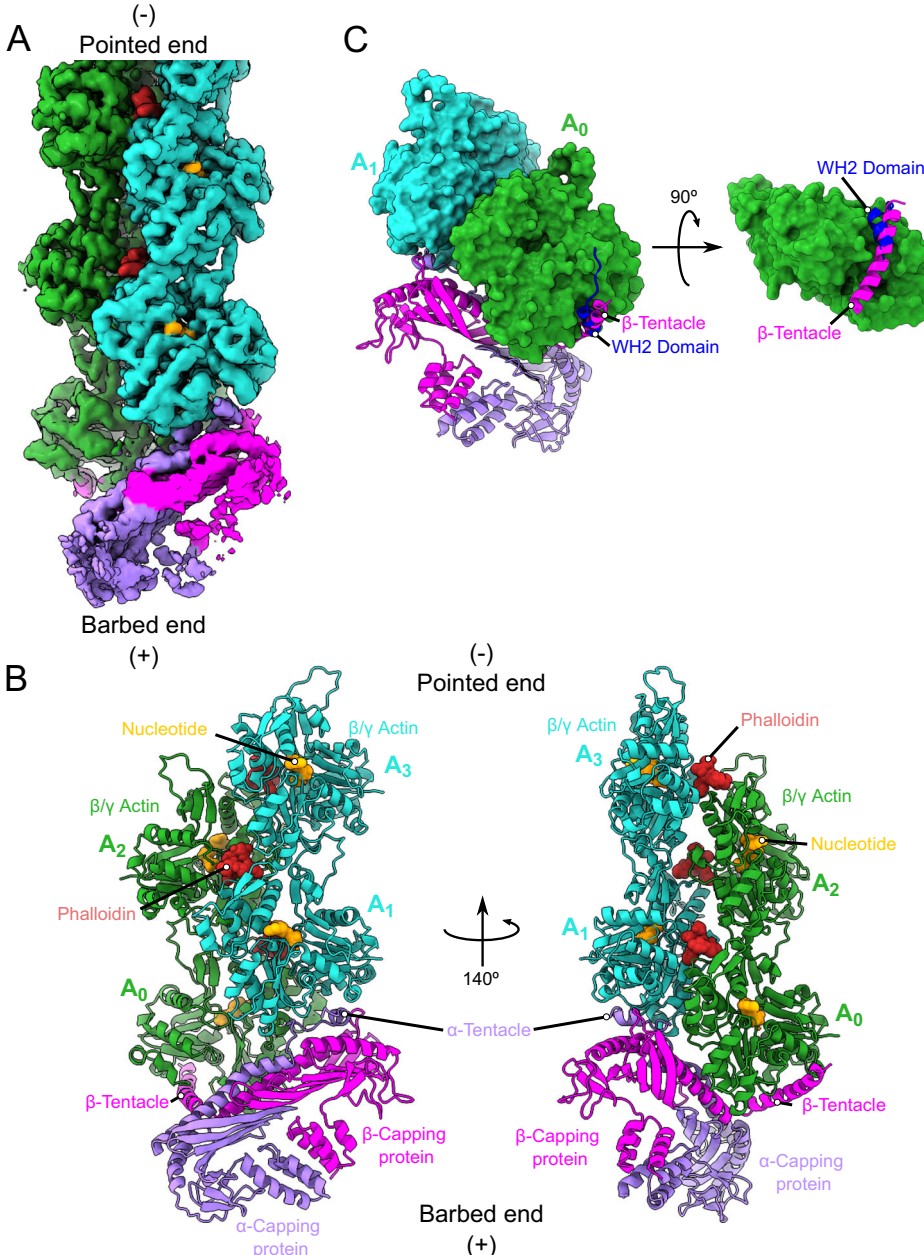

**Fig. 2 Cryo-EM structure of CP-bound barbed ends. A** Overview of the Cryo-EM map of capped F-actin. The density has been sharpened and denoised using deepEMhancer[82] **B** Atomic model of CP bound to the barbed end. While the penultimate protomer interacts extensively with CP, the barbed end side of the terminal protomer is bound mostly by the β tentacle. We highlight the position of the bound small molecules. Actin subunits are labeled $A_0$ to $A_3$ starting from the terminal actin protomer. **C** The structure of WASP-bound G-actin[51] (PDBID:2A3Z) was superimposed onto the terminal protomer of the complex. The β tentacle occupies the binding pocket that a WH2 domain would need to bind the terminal protomer. Actin monomers are shown as surface, while the tentacle and the WH2 helices are shown as cartoons.

deeply wedged in a hydrophobic cleft at the barbed end of this subunit. In contrast, CP makes few contacts with the barbed end face of the ultimate protomer (Supplementary Fig. 3). Contacts between CP and the terminal subunit are dominantly formed between CP's core and the inner side of the ultimate protomer. However, a clear density occupies the hydrophobic cleft at the barbed end of this subunit, which can be unequivocally assigned to the β tentacle (Fig. 2B, C). The density in this region was slightly weaker, but comparable to the body of the terminal actin protomer (Fig. 2A), suggesting that the β tentacle remains mostly docked to actin but retains some flexibility. The placement of the β tentacle has only been speculated from a previously reported low-resolution structure of capped skeletal actin[27] and agrees

with prior computational models and mutational analysis[28,29]. As predicted previously[28], this secondary barbed end binding site conspicuously overlaps with and mirrors the interaction between WH2 domain-containing proteins of the WASP family and actin monomers (Fig. 2C). The strong conservation of the β tentacle among even evolutionarily distant CP family members suggests that this specific type of interaction serves an important function.

**Deletion of the CP β tentacle modestly affects barbed end capping kinetics.** To understand the role of this conserved secondary actin-binding site in CP, we removed the last 23 amino acids of the beta subunit (CPαβΔ23) comprising the full β tentacle

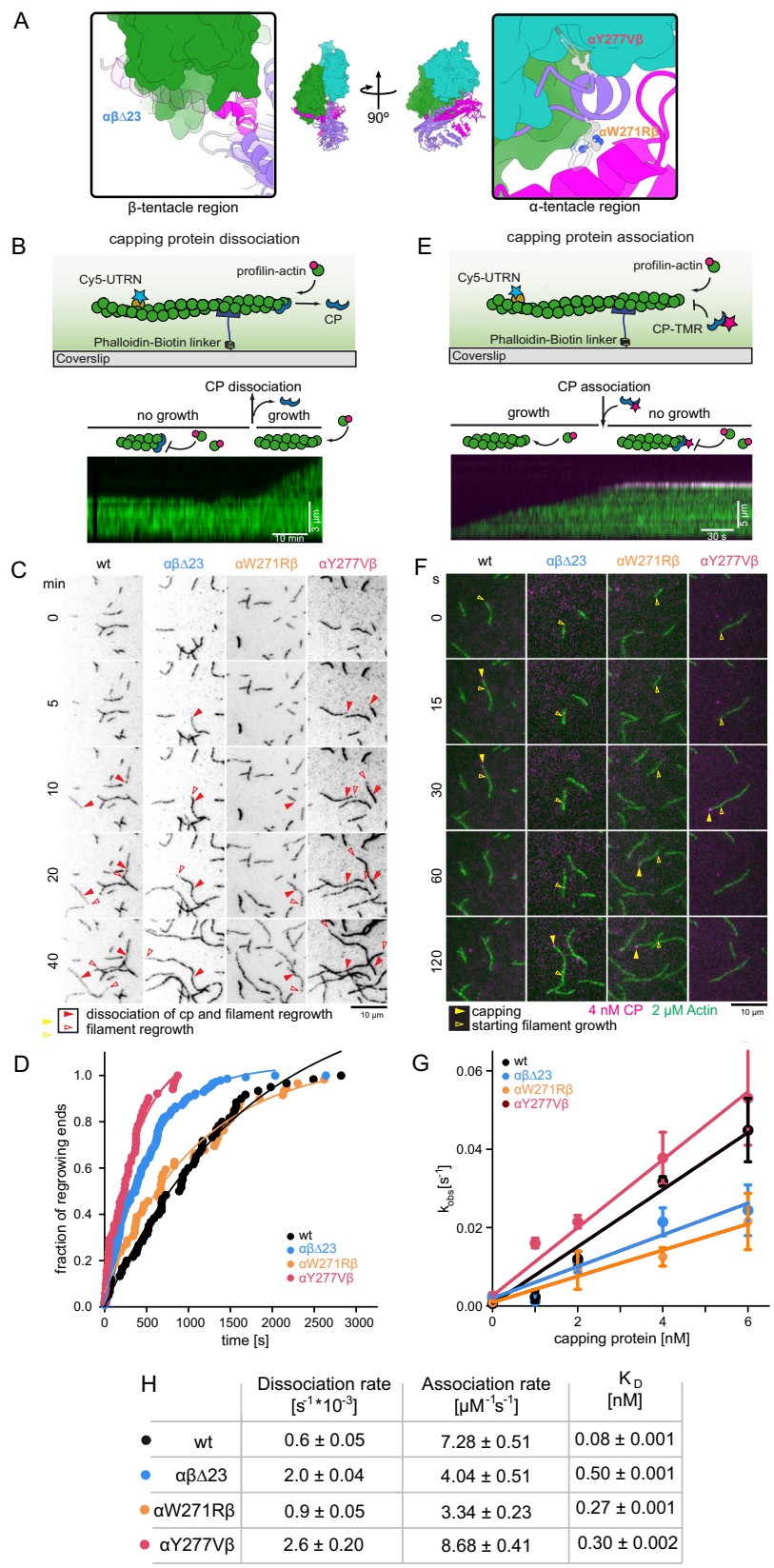

(Fig. 3A) and studied the effect on filament barbed end association and dissociation kinetics by TIRFM assays in vitro (Fig. 3). First, we measured the rate of CP dissociation from growing, surface-tethered actin filaments at the single filament level (Fig. 3B, C). Due to the extraordinarily long lifetime of CP at the barbed end (>10 min), we could not visualize single CP molecules

directly, but inferred dissociation indirectly from the moment of filament regrowth (Fig. 3B, C). Mono-exponential fits to the dwell time courses yielded the dissociation rate constants ($k_{off}$, Fig. 3D). In a second line of experiments, we probed CP association to the barbed ends of surface-tethered filaments by dual-color single molecule assays (Fig. 3E, F). Observed capping rates

**Fig. 3 The CP β tentacle is dispensable for capping of single actin filaments. A** Overview of the location of CP mutations. Colors are as in Fig. 2. Insets highlight the region of the β tentacle deleted (green), and the color scale on the actin surface depicts increasing hydrophobicity from gray to white to yellow. **B** Scheme of actin filament barbed end re-growth after dissociation of CP via TIRFM (top). Single capped filaments were visualized by Cy5-UTRN261N (10 nM, green) in the presence of profilin-actin (2 μM) and myotrophin/V1 (20 nM) after CP washout. CP dissociation leads to growth of the filament visualized in the kymograph (bottom). Over 50 such kymographs were analyzed per experiment, which was repeated independently three times with similar results. **C** TIRFM time lapse images of filaments following the washout of either wt CP or mutants, as indicated. Conditions are as in **B**. Filled red arrowheads indicate the positions of barbed ends at the moment of CP dissociation in each case, and empty red arrowheads follow the positions of (re-) growing barbed ends. The experiment was repeated three times with similar results. **D** Time traces of the fraction of uncapped barbed ends after washout of either wt CP or mutants, as indicated, at $t = 0$. Dissociation rate constants ($k_{off}$) were derived from mono-exponential fits (see Methods). Source data are provided as a Source Data file. **E** Scheme of single molecule TIRFM assays (top). Individual surface-attached filaments visualized by Cy5-UTRN261N (10 nM, green) grow from profilin-actin (2 μM) in the presence of TMR-CP (4 nM, magenta). CP association results in the termination of filament growth as visualized in the kymograph (bottom). Over 50 such kymographs were analyzed per experiment, which was repeated independently three times with similar results. **F** TIRFM time lapse images of filament elongation (green) in presence of wt or mutant capping protein (magenta) as indicated under conditions as in **E**. Empty yellow arrowheads indicate initial positions of barbed ends and filled yellow arrowheads the positions at the moment of CP association. The experiment was repeated three times for each CP concentration with similar results. **G** Observed reaction rates ($k_{obs}$) for capping protein wt and mutants as a function of CP concentration (Supplementary Fig. 4). The observed reaction rates ($k_{obs}$) were derived from the mean of $N = 3$ independent experiments for each CP concentration. Plotted data also contains the independently measured dissociation rates as y-intercepts. Association rate constants ($k_{on}$) are calculated from linear fits to the data with the y-intercepts fixed to the measured off-rates. Error bars are SD. Source data are provided as a Source Data file. **H** Summary table of dissociation ($k_{off}$), association rate constants ($k_{on}$) and equilibrium dissociation constants ($K_D$, calculated from the former rate constants) for the interaction of CP (wt or mutants as indicated) with actin filament barbed ends. Errors are SD of the mean values from 3 independent experiments ($k_{off}$) or the SEM of the linear fit ($k_{on}$).

(Supplementary Fig. 4) were plotted against the total CP concentration, and linear fits to the data, constrained by the independently measured off-rate, yielded association rate constants ($k_{on}$, Fig. 3G). In overall agreement with prior bulk measurements[29,38], we found that the deletion of the entire CP β tentacle reduced the affinity for the barbed end only by less than one order of magnitude. This was the result of moderate changes in the association (1.8-fold) and dissociation (8-fold) rate constants. The moderate drop in affinity suggests that engagement of this secondary binding site is not essential for filament capping.

The dwell time of CP at filament ends in vitro exceeds its lifetime in cellular actin networks by two orders of magnitude[39–41]. This is in part the result of its active removal from filaments ends via twinfilin and potentially other allosteric regulators[2,42,43]. It is reasonable to assume therefore that its association rate should be functionally more relevant compared to its exceedingly slow in vitro dissociation rate. Even in the absence of its β tentacle, CP still dwells at filament ends far longer than it should be necessary to carry out its capping function at the leading edge of motile cells. To control for the moderate contribution of the β tentacle to barbed end capping kinetics and to uncouple this effect from a potential unrelated secondary function, we designed weakening mutants distant from β tentacle within the primary barbed end binding site of CP (Fig. 3A, see Methods). We identified two single site mutants within the CP α subunit (αW271Rβ and αY277Vβ) that altered the thermodynamics and kinetics of barbed end binding to a similar degree as β tentacle removal (Fig. 3D, G, H). We therefore refer to these two variants as "kinetics-mimicking" mutants that cause effects on single filament capping rates closely resembling those seen after β tentacle deletion.

**Deletion of the CP β tentacle drastically inhibits reconstituted branched actin network assembly.** Because the deletion of the CP β tentacle results in only mild changes in the kinetics of capping, we wondered how this relates to its potentially more complex role in regulating branched actin network dynamics. To address this question, we assembled branched actin networks from polystyrene beads that were coated with a minimal Arp2/3-activating fragment of the NPF protein WAVE1 (WAVEΔN[44]). We initiated network assembly by the addition of Arp2/3 complex (100 nM), CP (wildtype or mutants, 100 nM), and 1:1

complexes of profilin–actin prepared by size exclusion chromatography (5 μM, see Methods[45]); conditions previously shown to promote rapid and sustained actin network growth[5,8]. The tight affinity of cytoplasmic actin together with isolation of stoichiometric profilin-actin complexes reduces the free actin monomer concentrations in our assays sufficiently (to about 290 nM) to prevent significant nucleation of filaments in solution[8,45]. To visualize the kinetics of all biochemical processes contributing to the assembly of the actin network—actin polymerization, Arp2/3-driven nucleation and CP-mediated capping—we included a fluorescently-labeled filament binding probe (Alexa488-lifeact[45,46]) and used fractions of Arp2/3 complex and CP labeled with matching fluorescent dyes (Fig. 4A). We kinetically arrested network assembly very soon (4 min) after the initiation of growth by adding a 3-fold molar excess of Latrunculin B and phalloidin, which stabilizes existing actin filaments and prevents the nucleation of new ones[5]. Upon kinetic arrest, we immediately proceeded to multi-color imaging of formed actin networks (Fig. 4B).

We observed that reactions with CP lacking the β tentacle generated actin networks with drastically altered assembly dynamics compared to reactions with wildtype CP (Fig. 4B, top). Averaged intensity profiles showed that networks grown in the presence of CPαβΔ23 contained less polymerized actin and drastically reduced Arp2/3 complex and CP amounts (Fig. 4B, bottom). Relative to the wildtype CP control, these networks grew with about 3-fold reduced average velocities (Fig. 4C). To directly quantify the rates at which actin, CP, and the Arp2/3 complex join the growing network at the NPF-coated surface, we integrated the fluorescence intensity of each component in individual actin networks and divided these values by the reaction time (Fig. 4D). This analysis revealed that the deletion of the CP β tentacle dramatically (20-fold, Supplementary Table 2) reduced network incorporation rate of CP much more than predicted by its modest defect in capping at the single molecule level (Fig. 3H). Conversely, we also observed a drastic reduction in the rate of Arp2/3 complex network incorporation (Fig. 4D). To test whether this reduced filament nucleation rate at the network level was simply a consequence of its modestly reduced capping rate, we turned to the CP mutants kinetically mimicking the β tentacle deletion (αW271Rβ and αY277Vβ). Decisively, we observed that both mutants were capable of generating branched actin networks

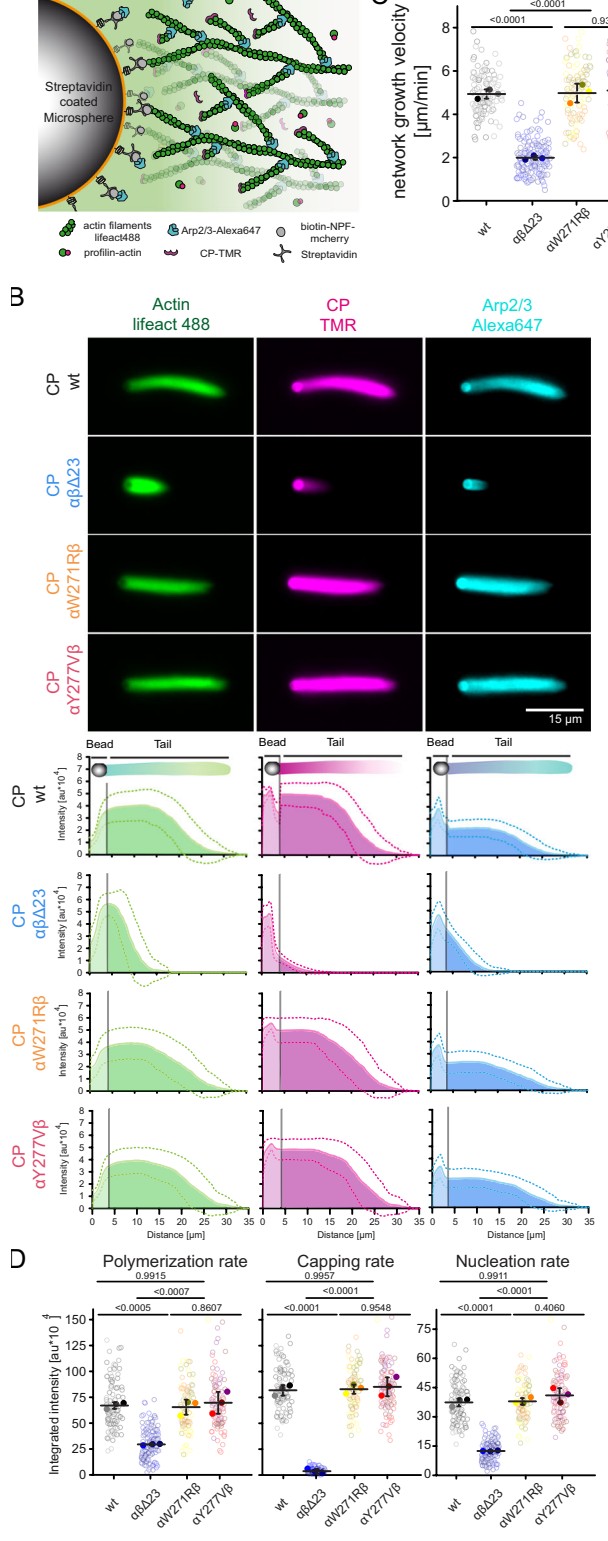

**Fig. 4 The CP β tentacle is essential for reconstituted branched actin network assembly. A** Scheme describing bead-motility-assay. Dendritic actin networks are assembled from polystyrene microspheres (Ø = 3 μm) coated with WAVE1ΔN via biotin-streptavidin linkages. Dense branched actin networks assemble from these beads and incorporation of all constituting components can be visualized via matching fluorescent labels as indicated. **B** Top: Representative epifluorescence images of dendritic actin networks grown from WAVE1ΔN-coated microspheres using 5 μM profilin–actin complexes (see Methods), 100 nM CP (either wt or mutants as indicated, 20% TMR-labeled), 100 nM Arp2/3 (20%-Alexa647 labeled). Reactions were kinetically arrested after 4 min using phalloidin (15 μM), Latrunculin-B (15 μM) and Alexa488-lifeact (15 nM) to visualize filamentous actin. The experiment was repeated three times with similar results. Bottom: Quantification of the averaged intensity profiles for the indicated dendritic network components. Source data are provided as a Source Data file. **C** Plots of network growth velocities of dendritic actin networks grown in presence of either wt or mutant CP. Source data are provided as a Source Data file. **D** Plots of polymerization, capping and nucleation rates of dendritic actin networks grown in presence of either wt or mutant CP (see Methods). Source data are provided as a Source Data file. Quantifications in **C** and **D** were done for $n = 25$ actin networks from $N = 3$ independent experiments, error indicator = SD of the mean. $P$-values were derived from one-way ANOVA Tukey tests.

therefore repeated our bead motility experiments by adding an excess of either profilin (500 nM) or thymosin-$β_4$ (4.5 μM) to 5 μM profilin-actin. According to mass action, either addition will reduce the free actin monomer concentration by about 50% (from 290 to 140 nM). Importantly, we observed that the selective defects in branched network assembly resulting from the CP beta tentacle deletion are preserved in either condition (Supplementary Fig. 5). We therefore deduced that these defects are robust towards changes in the actin monomer pool and can be similarly observed at various levels of free actin monomers. In conclusion, we found that the strongly diminished rates of network incorporation of both CP and Arp2/3 complex resulting from the CP β tentacle deletion were not caused by its modestly altered capping rates observed at the single filament level. Instead, the data indicates that the CP β tentacle serves an important secondary function in branched network assembly not directly related to the termination of actin filament growth.

**Deletion of the CP β tentacle inhibits Arp2/3-dependent branching by tethering NPF proteins to capped filament ends through their WH2 domain.** The CP β tentacle interacts with the central binding cleft of the terminal actin protomer at the barbed end in a manner that sterically prevents binding of other WH2 domain-containing proteins such as NPFs (Fig. 2C). We therefore hypothesized that capping protein controls the rate of Arp2/3-mediated branching at the level of the NPF as part of a feedback mechanism (Fig. 5A): CP occupies both WH2 binding sites at the penultimate and terminal barbed end subunits, effectively masking the capped filament end from NPF binding. Loss of the β tentacle liberates one of these binding sites, which might drive the sequestration of proximal NPF WH2 domains. This type of tethering should therefore reduce the amount of NPF molecules loaded with actin monomers ready to activate the Arp2/3 complex.

To test this hypothesis, we employed a recently developed Förster resonance energy transfer (FRET) method to directly measure partitioning of WH2 domains between actin monomers and filament ends[44] (Fig. 5A, Supplementary Fig. 6). To this end, we conjugated a fluorescent donor (Alexa488) to a site immediately upstream of the WH2 domain of WAVE1 and

with assembly kinetics nearly indistinguishable from wildtype CP (Fig. 4B-D).

Because we reconstituted branched network assembly at near-saturating levels of profilin, which reduces the availability of free actin monomers, we were wondering whether our observations depended on the exact composition of the soluble actin pool. We

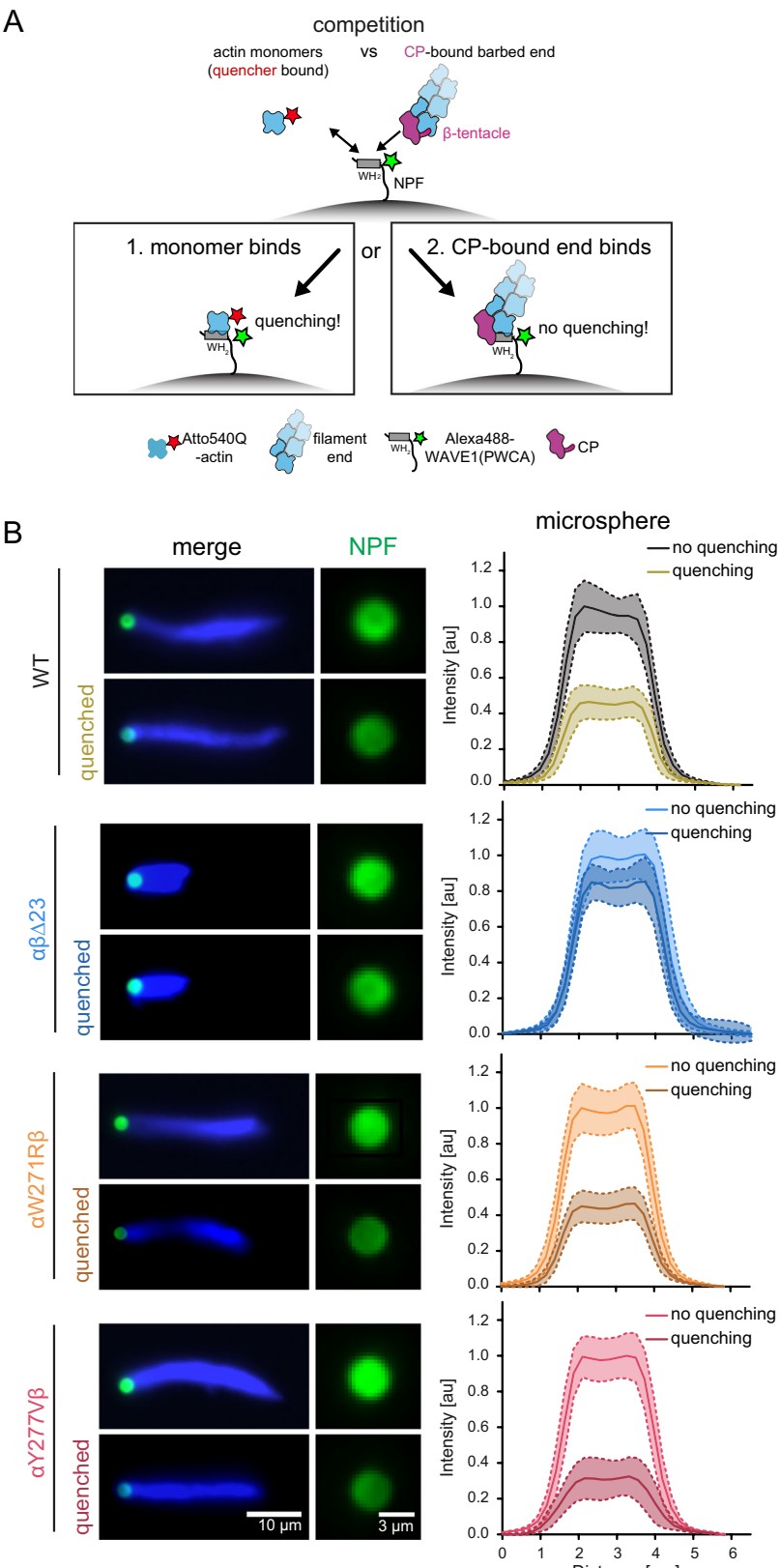

conjugated a non-fluorescent quencher (Atto540Q) to monomeric actin in a manner that does not affect their binding affinities[44] (see Methods). We then immobilized this labeled WAVE1 fragment on polystyrene beads and assembled branched networks in the presence of either wildtype CP or mutants. We added quencher-labeled and Latrunculin B-stabilized actin

monomers at the moment of kinetic network arrest through Latrunculin B, phalloidin and myotrophin/V-1. The latter inhibitor preserves free barbed ends[8]. To minimize the impact of CP dissociation from capped ends after network arrest, we immediately proceeded to imaging and did not analyze NPF-coated beads longer than 5 minutes after inhibitor and

**Fig. 5 Loss of the CP β tentacle tethers NPFs to capped filament ends via their WH2 actin binding site. A** Scheme of the FRET setup. Surface-bound WAVEΔN molecules are donor-(Alexa488-) labeled upstream of their WH2 site. NPF WH2 can either interact with quencher- (Atto540Q-) labeled actin monomers resulting in a decrease of donor fluorescence or unlabeled terminal protomers of filament barbed ends, resulting in no change in fluorescence. Terminal protomers are unlabeled, since quencher-labeled monomers are introduced only upon network arrest. Loss of the β tentacle vacates the WH2 binding site at the terminal protomer of capped ends, tipping the balance in favor of end-binding. **B** Left: Representative epifluorescence images of Alexa488-WAVEΔN-coated microspheres (green) and dendritic actin networks (visualized by Cy5-UTRN₍261₎, blue) 3 min after arrest before (top) and after (bottom) quenching with Atto540Q-labeled actin monomers. Protein concentrations were as in Fig. 4 in the presence of wt or mutant capping proteins, as indicated. Right: Average intensities (continuous lines at the center of the error band, dashed lines error = SD) of the NPF-fluorescent signal intensity on the microsphere surface in the presence or absence of Atto540Q-labeled actin monomers (n = 25 beads per condition). Source data are provided as a Source Data file.

quencher addition. Alexa488-WAVEΔN-coated beads in physical contact with actin networks assembled from either wildtype or kinetics-mimicking (αW271Rβ and αY277Vβ) CP mutants displayed similarly strong (by 59-69%) quenching of NPF donor fluorescence, indicating that a majority of NPF molecules were able to bind quencher-labeled actin monomers from solution (Fig. 5A, B). Strikingly, quenching was substantially diminished (to 14%) for Alexa488-WAVEΔN beads tethered to actin networks grown in the presence of CP lacking its β tentacle (Fig. 5B, Supplementary Fig. 6). These results verify that the loss of the β tentacle locks the majority of NPF molecules in an inactive state tethered to capped filament ends. In summary, this strongly suggests that the conservation of this secondary end binding site in CP arises from the necessity to mask the terminal actin subunit at the barbed end from WH2 binding.

Inspired by this result, we wondered whether CP exerts a similar function in blocking the NPF binding site also at the penultimate actin subunit through its α tentacle. If this were the case, removing the α tentacle could vacate this second NPF binding site, which might cause defects similar to β tentacle deletion. Previous work established that truncation of the entire α tentacle dramatically inhibits barbed end binding and therefore cannot be used to test this hypothesis[29]. Our structure, however, suggested that removing the very end of the α tentacle by only a few amino acids might liberate the NPF WH2 binding site at the penultimate subunit (Supplementary Fig. 7). We therefore created two additional CP variants that removed the last 8 or 9 residues of the α tentacle and studied the effect of either deletion in reconstituted branched network assembly (Supplementary Fig. 7). Decisively, both CP variants behaved like wildtype capping protein, indicating that they were active in barbed end binding as anticipated, but did not inhibit nucleation like the β tentacle deletion (Supplementary Fig. 7). We believe the latter finding reflects either of two possibilities: (i) The short deletions do not liberate the WH2 site at the penultimate subunit entirely. Indeed, the NPF binding site of this subunit comes very close to the CP body (Supplementary Fig. 7). Moreover, the WH2 domain represent only a fraction of the NPF, suggesting possible clashes with CP. (ii) Ultimate and penultimate actin subunits are differentially conducive to NPF WH2 binding. While we do not observe obvious structural differences between these subunits, the penultimate subunit will be more distant (by about 2.5 nm) from the surface-immobilized NPF on average. This in itself might lead to preferential NPF binding to the last subunit, explaining why the α and β tentacles of CP are not functionally equivalent.

**The β tentacle is essential for CP localization to lamellipodial actin networks and efficient leading edge protrusion.** To determine how our in vitro results translate to the function of CP in regulating branched actin network dynamics in mammalian cells, we generated engineered murine B16-F1 cell lines, in which expression of either one of the two CP subunits was genetically disrupted by CRISPR/Cas9-mediated genome editing. We first stably knocked out the single gene (*CapZb*) from which all three known CPβ isoforms are derived[47]. This successfully resulted in a clonal cell line devoid of detectable CPβ expression (CapZβ KO#10, Fig. 6A). In parallel, we attempted to inactivate both genes (*CapZa1 and −2*) responsible for the expression of the two ubiquitous CPα isoforms[1,48]. The murine genome contains a third CPα gene (CapZa3), however the resulting protein is more divergent and only expressed in the male germ cell lineage[1,49]. Simultaneous, CRISPR/Cas9-mediated targeting of *CapZa1* and −2 did not yield clones entirely devoid of CPα, indicating that the CPα subunit might be essential for cell viability. Nonetheless, we were able to select a clone resulting from synchronous targeting of *CapZa1* and -2 that showed markedly reduced total CPα expression levels, with the CPα1 isoform being virtually absent (Fig. 6A).

In agreement with prior knockdown/knockout studies[22,23,39], functional interference with either CP subunit caused consistent defects in leading edge dynamics reflected by significantly altered morphologies at the peripheries of these cells compared to wild type (Fig. 6B, C). These phenotypes coincided with strongly compromised protrusion velocities in each genotype (by app. 70%), which could be largely rescued by EGFP-tagged wildtype variants of respective, disrupted gene (Fig. 6C, D). Inefficient protrusion observed in CRISPR/Cas9-treated clones derived from their leading edge membranes undergoing rapid fluctuations, strongly contrasted by the smooth, homogeneous protrusion seen upon rescue with wildtype CPβ and −α as well as the kinetics-mimicking mutants of the latter (αW271Rβ or αY277Vβ, Fig. 6D bottom right) when expressed at similar levels (Supplementary Fig. 8). Furthermore, the restoration of protrusion velocity was similarly efficient with the latter CPα-mutants as seen with wild type CPα (Fig. 6D, right panel). As only exception, expression of CPβ lacking its tentacle region in the CPβ-deficient clone was significantly less potent in its rescue efficiency compared to wildtype CPβ when expressed at comparable levels (Supplementary Fig. 8), with a protrusion efficiency found in the latter condition statistically indistinguishable from untransfected KO cells (Fig. 6D, left panel). The functional difference between CP lacking its β tentacle (αβΔ23) and the corresponding kinetics-mimicking mutants (αW271Rβ or αY277Vβ) also correlated with their accumulation in restored protrusions, since all EGFP-tagged constructs accumulated robustly at protruding cell edges except for αβΔ23 (Fig. 6C). Importantly, we confirmed that the levels of both, lamellipodial F-actin and Arp2/3 complex were substantially reduced in the absence of the beta tentacle, as expected (for representative images see Supplementary Fig. 8), albeit not abolished. These effects concur with our model and closely resemble the observations obtained with our reconstitutions, in which we saw a similarly strong reduction of both factors in branched networks in vitro. We therefore concluded that the strongly diminished accumulation of β tentacle-truncated CP (αβΔ23) in lamellipodial networks at least partly coincides with

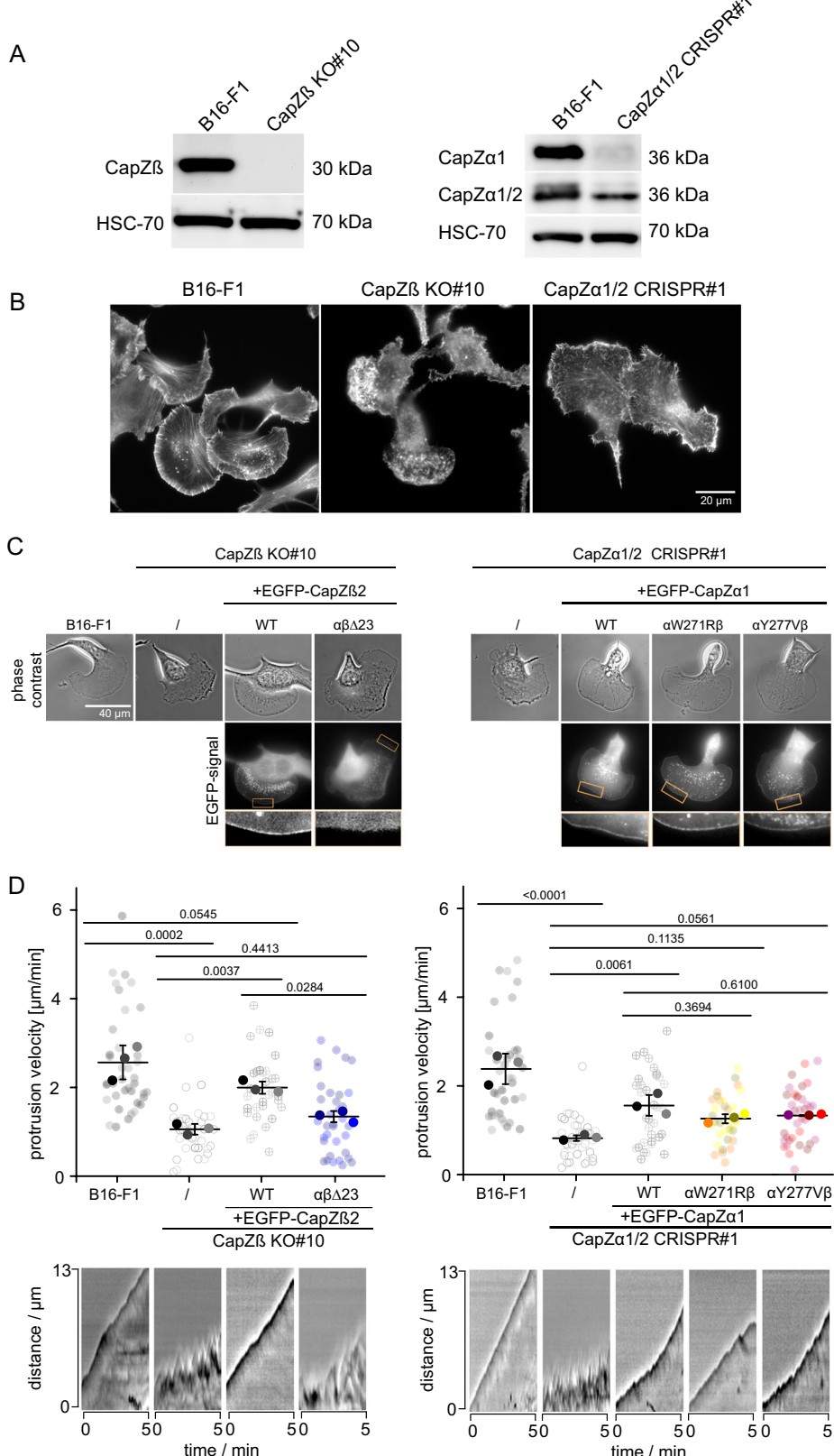

reduced Arp2/3 complex-mediated branching activity in those cells. Taken together, these data show that the CP β tentacle serves an important function in regulating lamellipodial actin dynamics secondary to simply contributing to high-affinity barbed end binding of actin filaments.

## Discussion

Combining structural and cell biology with in vitro reconstitution, we have elucidated a mechanism that links the two central antagonistic biochemical reactions –capping and nucleation- in the assembly of branched actin networks. Our results identify the WH2 domain of the NPF as the central coupling element that can

**Fig. 6 The β tentacle is essential for CP localization to lamellipodial actin networks and efficient leading edge protrusion. A** Lysates of B16-F1 control cells, cells with CapZβ KO (clone #10, left) and CapZα1/2 CRISPR/Cas9-treated cells (clone #1, right) subjected to western blotting using CapZβ (left) or CapZα1 and CapZα1/2 (right) antibodies. The experiment was performed once. **B** Cell morphologies of B16-F1 *versus* CapZβ KO and CapZα1/2 CRISPR cells stained for the actin cytoskeleton with phalloidin. The experiment was performed once. **C** Representative phase contrast and fluorescence live cell imaging frames of B16-F1 control cells, CapZβ KO and CapZα1/2 CRISPR/Cas9-treated B16-F1 cells with or without expression of EGFP-tagged CapZβ2 (left) or CapZα1 (right) variants, as indicated. Boxed regions (orange) highlight insets at the bottom to reveal localization patterns of respective, EGFP-tagged rescue constructs. The experiment was repeated three times with similar results. **D** Quantification of protrusion velocity (with representative kymographs at the bottom) of B16-F1 control cells, CapZβ KO (clone 10, left) or CapZα1/2 CRISPR clone 1 (right) cells and the latter expressing CapZβ2 or CapZα1 variants as indicated. Quantifications were done for $n = 32$ cells from $N = 3$ independent experiments, error indicator = SD of the mean. P-values were derived from one-way ANOVA Tukey tests. Source data are provided as a Source Data file.

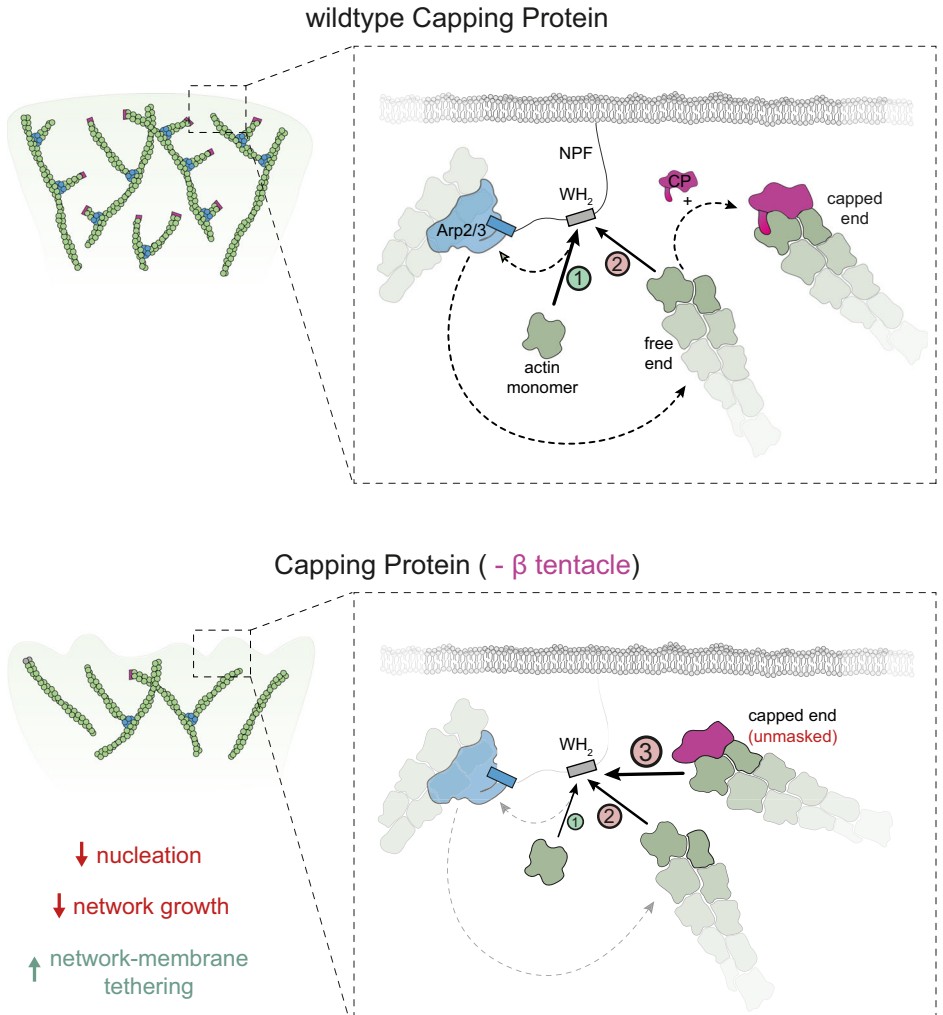

**Fig. 7 Scheme of the barbed end interference model.** Top panel: Actin monomers compete with free barbed ends for NPF WH2 binding. Binding of the former generates the nucleation-competent NPF state (1), whereas binding of the latter sequesters the NPF in an inactive configuration (2). Binding of wildtype capping protein masks the terminal subunits preventing NPF interaction, thereby indirectly stimulating nucleation. Bottom panel: Deletion of the β tentacle unmasks the terminal actin protomer, allowing capped ends to tether additional NPF proteins in an inactive state (3). This strengthens the tethering between actin network and the membrane and decreases the nucleation rate and network growth velocity.

assume distinct states (Fig. 7): (1) Bound to an actin monomer ready to activate the Arp2/3 complex and (2) tethered to a free filament barbed end in a nucleation-inactive configuration. We propose that the latter interaction provides negative feedback to Arp2/3 complex activity and refer to this mechanism as barbed end interference. Importantly, both free, polymerizing barbed ends and capped ends unmasked by the CP β tentacle deletion are capable of sequestering WH2 domains. In the absence of the CP β tentacle, WH2 domains cannot discriminate between capped and

free barbed filament ends, effectively creating an additional third state that tethers and inactivates the NPF (Fig. 7). We propose that this results in excessive NPF WH2 sequestration and strong inhibition of Arp2/3-dependent branching in the context of the motile network.

Barbed end interference clearly differs from the previously proposed monomer gating hypothesis[5]. Monomer gating assumes a kinetic competition between free barbed ends and the Arp2/3 complex for a limiting "flux" of monomeric actin transiently

bound to the nucleation promoting factor. Barbed end interference on the other hand simply reflects a direct interaction between NPFs and filament barbed ends. Our results strongly argue for the latter, because a significant fraction of NPFs bind to actin filament ends even when network assembly is kinetically arrested i.e. in the total absence of monomer flux[50] (Fig. 5).

WH2 domains of WAVE-family proteins bind actin monomers more tightly ($K_D < 1\,\mu M$) than filament barbed ends ($K_D > 10\,\mu M$)[44,51,52]. How can barbed ends—either free and polymerizing or capped and "unmasked" by the loss of the CP β tentacle—compete with monomers for NPF binding given this large difference in affinity? Several factors might strongly tip the balance towards end binding. First, the density of polymerizing barbed ends at the active front of lamellipodial networks is extremely high (500–2000/$\mu m^2$ (refs. [5,8,19])). Second, in the context of a growing actin network, free ends are confined to the immediate proximity of the NPF, which is attached to the boundary they push against[8,19]. This type of dimensionality reduction might greatly favor association. Third, bare actin monomers unlikely exist at levels significantly above their critical concentration under physiological conditions ($c_c < 1\,\mu M$). Indeed, the majority of soluble actin is bound to profilin in living cells[45,53], which itself competes with and reduces the monomer occupancy of NPF WH2 domains[44]. The exact contributions of each of these mechanisms to barbed end interference remains to be tested. Importantly, both profilin[44,54] and filament barbed ends can indirectly reduce NPF activity through competition with WH2 for monomer binding. While we presently do not fully understand how these two distinct effects combine in the cellular environment, our results show that barbed end interference must constitute an important layer in the control of NPF activity in vivo. In the absence of such an indirect negative feedback, capping and nucleation within branched networks are not tightly coupled, as observed both in cells and in reconstitutions.

In addition to resolving the stimulating effect of CP on Arp2/3-dependent nucleation, barbed end interference explains a multitude of observations otherwise hard to rationalize. Most importantly, it explains why Arp2/3 complex activation by WASP-family proteins requires the transfer of an actin monomer bound to the NPF WH2 domain. The recent discovery of a second class of Arp2/3 complex activators devoid of an actin monomer binding site[55,56] demonstrates that monomer delivery is not a mandatory step in nucleation[32]. Instead, we propose that the NPF WH2 domain evolved as an important, conserved feedback element that keeps autocatalytic branching at bay[24]. Barbed end interference also provides an explanation for the universal conservation of the CP β tentacle. Previous genetic and biochemical work indicated that this secondary end-binding site might be dispensable for capping and contributes little to the termination of filament growth as such[29,38]. We show here that it nonetheless serves a critical function in branched network assembly, which is to prevent the capped filament end from WH2 binding. All of these points highlight that the proteins constituting the core branched actin network motor—Arp2/3 complex, CP and NPFs—are more intimately linked than previously appreciated and have evolved as a functional unit. This illustrates the necessity to study them not only individually as done traditionally, but rather collectively under realistic boundary conditions when constructing a force-generating network.

Several actin-binding proteins unrelated to CP such as gelsolin-family members or epidermal growth factor receptor kinase substrate 8 (EPS8) possess barbed end capping activity[1,2]. However, these proteins are less conserved within the eukaryotic domain and display a more limited expression across distinct cell types and tissues. Whether their inability to compensate for the loss of CP is the result of a distinct mode of binding at the filament barbed end that does not result in competition with NPF

WH2 domains remains to be investigated. Competition for overlapping binding sites at the exposed face of terminal barbed end subunits likely represents a general mechanism by which barbed end-binding proteins reciprocally control their activity. For instance, twinfilin likely removes CP from the barbed end by competition with the tentacles of both CP subunits for actin binding[42,43]. We speculate that deletion of the CP β tentacle might thus not only favor NPF binding as shown here, but also accelerate twinfilin binding and uncapping, which remains to be tested. Such rapid uncapping might also contribute to the impaired localization of CPαβΔ23 to lamellipodial actin networks. Aside from direct CP regulators, formins can also accelerate barbed end uncapping[57,58]. Although structural details are not known, it is likely that this also occurs by competition for partially overlapping sites involving the exposed central barbed end cleft. Finally, we have recently shown that formins accelerate the release of profilin from the barbed end they processively elongate[45]. Whether this is the result of direct competition or due to allosteric communication remains to be resolved. The structural approach established here provides a powerful strategy to study how all of these and other unrelated end-binding proteins target actin filament ends.

## Methods

### Protein biochemistry—purification and labeling

*Native (β, γ) -actin.* Native bovine (β, γ)—actin was purified from bovine thymus tissue according to the methods described previously[45]. Briefly, (β, γ)—actin was purified from fresh bovine thymus tissue. After lysis of the thymus tissue, the solution was hard spun and filtered to generate a cleared supernatant that was incubated with His$_{10}$-gelsolinG4-6 fragment to promote the formation of actin:-gelsolin G4-6 complexes. Next, the lysate was circulated over a Ni$^{2+}$ superflow column. Actin monomers were eluted and polymerized into filaments. After ultracentrifugation, actin filaments were resuspended in F-buffer (1xKMEI, 1xBufferA) and stored in continuous dialysis at 4 °C. F-buffer containing fresh ATP and TCEP was continuously exchanged every 4 weeks. Actin was depolymerized through dialysis into BufferA (2 mM Tris, 0.2 mM ATP, 0.1 mM CaCl$_2$, 0.5 mM TCEP) for 6 days and gelfiltered over a Superdex 200 16/600 column prior to all experiments.

*Profilin and profilin-actin complexes.* Human profilin1 and stoichiometric complexes of profilin and β, γ—actin were purified as described[44,45]. Briefly, human profilin1 was expressed as untagged protein in *E. coli* Rosetta cells at 30 °C for 4.5 hr. Profilin1 was purified by ammonium sulfate precipitation, followed by ion-exchange (DEAE) and hydroxylapatite (HA) chromatography steps, followed by size exclusion chromatography (Superdex 200 16/600) into storage buffer (20 mM Tris-Cl pH 7.5, 50 mM NaCl, 0.5 mM TCEP). Proteins were snap-frozen in liquid nitrogen upon addition of 20% glycerol to the storage buffer, and stored at −80 °C.

To generate profilin-actin complexes, freshly gelfiltered actin monomers were incubated with 1.5x molar excess of profilin1 at 4 °C overnight. Next, profilin-actin complex was separated from free profilin by gelfiltration over a Superdex 200 10/ 300 GL into Buffer A. The complex was concentrated to working concentrations between 100 and 200 μM and stored at 4 °C up to two weeks without inducing nucleation.

*Atto540Q-quencher labeled actin.* For the labeling of actin monomers with Atto540Q-NHS, the complex of human profilin1 and *Acanthamoeba castellani* actin[59] was generated in Buffer A using a 2× molar excess of profilin1. Following gelfiltration into labeling buffer (2 mM Hepes pH 8.0, 0.1 mM CaCl$_2$, 0.2 mM ATP, 0.5 mM TCEP), the profilin-actin complex was labeled at its reactive lysine residues by incubating with a 3-fold excess of Atto540Q-NHS for 40 min at 23 °C. The reaction was terminated by adding Tris-Cl (pH 8.0) to a final concentration of 2 mM to the reaction mix. After 10 min incubation, soluble dye and protein were separated from solid particles by ultracentrifugation for 10 min at 300,000 *g*. Next, polymerization of actin was initiated by adding KMEI-buffer (10 mM imidazole pH 7.0, 50 mM KCl, 1.5 mM MgCl$_2$, 1 mM EGTA) and 1% of pre-polymerized, unlabeled, freshly-sheared actin filaments to the labeling mix[8]. After 1 hr of polymerization at 23 °C, filaments were separated from excess soluble dye, non-polymerizable actin and profilin by ultracentrifugation for 10 min at 300.000xg. Actin filaments were stored in F-buffer dialysis at 4 °C and dialyzed into Buffer A (for 6 days) prior to experiment. The degree of labeling (73–87%) was determined by absorbance at 280 nm and 543 nm.

*Myotrophin/V1.* Human myotrophin was expressed in *E. coli* BL21 Rosetta cells from a pETM11 vector containing an N-terminal 10xhis tag followed by a

TEV-cleavage site. After protein expression for 16 hrs at 18 °C, cells were lysed (50 mM KP$_i$ pH 7.3, 400 mM NaCl, 10 mM imidazole, 1 mM β-mercaptoethanol, 1 mM PMSF, DNaseI). The lysate was hard spun and purified by IMAC over a 5 mL HiTrap Chelating column. After gradient elution (50 mM KP$_i$ pH 7.3, 400 mM NaCl, 10 mM imidazole, 1 mM β-mercaptoethanol), the His$_{10}$-tag was cleaved by TEV protease overnight. After cleavage, the protein was desalted into lysis buffer and filtered over a HiTrap Chelating column. The flow-through was gelfiltered over a Superdex 200 column into storage buffer (20 mM Hepes pH 7.5, 50 mM KCl, 0.5 mM TCEP) and snap-frozen in liquid nitrogen supplemented with 20% glycerol. For long-term storage, the protein was stored at −80 °C.

*Capping protein.* Capping protein mutants were either generated via PCR amplification and Gibson cloning[60] (βΔ23) or via site-directed mutagenesis (αW271R, αY277V, EGFP-CapZb2). Primers sequences are listed in Supplementary Table 3. Murine heterodimeric capping protein (α1 in pETM20, β2 in pETM33[44]) were co-expressed in *E. coli* BL21 Rosetta cells. Following protein expression, cells were lysed (50 mM K$_2$PO$_4$ pH 7.3, 400 mM NaCl, 5 mM imidazole, 1 mM EDTA, 2 mM PMSF, 0.75 mM β-mercaptoethanol, 15 μg/mL benzamidine, DNaseI) and purified by IMAC over a 5 mL HiTrap Chelating column. After gradient elution (50 mM K$_2$PO$_4$ pH 7.3, 400 mM NaCl, 400 mM imidazole, 1 mM EDTA, 2 mM PMSF, 0.75 mM β-mercaptoethanol), the his-tag was cleaved by TEV/Precision proteases overnight. Following cleavage, the protein was desalted into low salt MonoQ buffer (10 mM Tris-Cl pH 8.0, 5 mM KCl, 1 mM EDTA, 2 mM PMSF, 1 mM DTT) and again filtered over a HiTrap Chelating column followed by a MonoQ run. After protein elution (10 mM Tris-Cl pH 8.0, 1 M KCl, 1 mM EDTA, 2 mM PMSF, 1 mM DTT) the protein was gelfiltered over a Superdex 200 column (10 mM Tris-Cl pH 7.5, 50 mM KCl, 0.5 mM TCEP, 20% glycerol), concentrated and either stored at −80 °C or directly transferred into a SNAP-labeling reaction.

To generate fluorescently-labeled capping protein, an N-terminal SNAP-tag[61] was fused to the beta subunit. For N-terminal SNAP-labeling, a 3x molar excess of SNAP Cell TMR-star was mixed with the protein and incubated for 4 hrs at 16 °C following an overnight incubation on ice. Next, the protein was gelfiltered over a Superose 6 10/300 GL column into storage buffer. The degree of labeling (50–70%) was determined by absorbance at 280 nm and 554 nm. The addition of N-terminal SNAP-tags did not affect the activity of capping protein as measured in single molecule/filament TIRF-M assays and bead motility experiments.

*Arp2/3 complex.* Native bovine Arp2/3 complex was purified from fresh bovine thymus by ammonium sulfate precipitation and ion exchange chromatography (DEAE, Source Q and Source S) followed by gelfiltration over a Superdex 200 as described[8,62]. The protein was either directly snap frozen with 20% glycerol and stored at −80 °C or transferred into a labeling reaction.

Arp2/3 complex was fluorescently labeled by the addition of 5-fold molar excess of maleimide-Alexa647 dye conjugate. After incubation on ice for 2 hrs, the reaction was terminated by adding 2 mM DTT for 30 min. Next, the complex was loaded onto an N-WASP-VCA column (5 mM Tris-Cl pH 8.0, 50 mM NaCl, 0.5 mM MgCl$_2$, 0.1 mM ATP, 2 mM DTT) and gradient-eluted over 100 column volumes (10 mM Tris-Cl pH 8.0, 1 M NaCl, 0.5 mM MgCl$_2$, 0.1 mM ATP, 2 mM DTT). Fractions containing the full complex were gel-filtered (5 mM Hepes pH 7.5, 50 mM KCl, 0.5 mM EGTA, 0.5 mM MgCl$_2$, 0.1 mM ATP, 0.2 mM TCEP) over a Superose 6 column, concentrated, snap frozen with 20% glycerol and stored at −80 °C.

*Preparation of an N-WASP-loaded column for Arp2/3 complex labeling.* Before immobilizing the N-WASP-VCA domain onto a 1 mL HiTrap NHS activated HP column, 10 mg of the protein were desalted into coupling buffer (50 mM Hepes pH 8.0, 500 mM NaCl, 0.5 mM DTT). The column was pretreated according to the recommendations of the manufacturer (GE healthcare). After all protein had bound to the resin, the column was equilibrated with wash buffer (5 mM Tris-Cl pH 8.0, 50 mM NaCl, 0.5 mM MgCl$_2$, 0.1 mM ATP, 2 mM DTT).

*N-WASP VCA.* Human N-WASP VCA was expressed with an N-terminal 10xhis-GST-tag using *E. coli* BL21 Star pRARE. After expression for 16 hrs at 18 °C, cells were lysed (100 mM KP$_i$ pH 7.2, 1 M NaCl, 0.5 mM β-mercaptoethanol, 5 mM imidazole, 1 mM PMSF, 15 μg/mL benzamidine, DNaseI) and protein was purified by IMAC over a 5 mL HiTrap Chelating column. After gradient elution, tags were removed by TEV protease cleavage overnight. After desalting the protein back into lysis/wash buffer, non-cleaved protein and free 10xhis tag was removed by passing the protein over a 5 mL HiTrap Chelating column. Next, the protein was loaded onto a MonoQ column (20 mM Tris-Cl pH 8.0, 50 mM NaCl, 1 mM DTT) and gradient eluted over 30 column volumes (20 mM Tris-Cl pH 8.0, 1 M NaCl, 1 mM DTT) followed by gelfiltration over a Superdex 200 column (20 mM Tris-Cl pH 8.0, 150 mM NaCl, 1 mM DTT). The protein was snap-frozen in liquid nitrogen with 20% glycerol and stored at −80 °C.

*WAVE WH2.* Human WAVE1 PVCA WH2 was expressed with an N-terminal 10xhis-ztag followed by penta-glycine and a C-terminal KCK motif in *E. coli* BL21 Star pRARE. After expression for 16 hrs at 18 °C, cells were lysed (50 mM Tris-Cl pH 8.0, 150 mM KCl, 0.5 mM β-mercaptoethanol, 1 mM PMSF, 15 μg/mL benzamidine, DNaseI), and protein was purified by IMAC over a 5 mL HiTrap

Chelating column. After gradient elution (50 mM Tris-Cl pH 8.0, 150 mM KCl, 0.5 mM β-mercaptoethanol, 300 mM imidazole), tags were removed by TEV protease cleavage overnight. After desalting the protein back into lysis/wash buffer, non-cleaved protein and free 10xhis tag was removed by passing the protein again over a 5 mL HiTrap Chelating column. Next, the protein was gelfiltered over a Superdex 75 column (10 mM Tris-Cl pH 7.5, 50 mM KCl, 0.5 mM TCEP). The protein was snap-frozen in liquid nitrogen with 20% glycerol and stored at −80 °C.

*NPF-WAVE-biotinylation by maleimide chemistry.* NPF (WAVE (PVCA)) molecules were labeled with EZ-Link maleimide-PEG$_4$-Biotin (Thermo Fisher Scientific) at the C-terminally located KCK, as recommended by the manufacturer. Briefly, NPF-KCK was desalted into reaction buffer (5 mM Hepes pH 7.5, 150 mM NaCl, 0.5 mM TCEP), mixed with 10x molar excess of EZ-Link maleimide-PEG$_4$-Biotin and incubated for 2 h on ice. After quenching the reaction with 1 mM DTT, the protein was gelfiltered over a Superdex 75 10/300 GL into storage buffer and directly transferred into another labeling reaction via sortagging.

*NPF-PEG-Biotin-Alexa 488/or mCherry by sortagging.* Following biotinylation, the NPF was transferred into a second labeling reaction by sortagging. The NPF was labeled with a LPETGG conjugated Alexa-488/or mCherry dye at the N-terminus as described[63]. Briefly, the NPF-protein was mixed with sortase (in a 3:1 molar ratio), Alexa488-LPETGG conjugate in 4 molar excess, 6 mM CaCl$_2$, 7.78 mM Tris pH 8.0, 150 mM KCl, 0.5 mM TCEP. After 18 hrs of incubation at 18 °C, the protein was separated from sortase and free dye by gelfiltration over a S200 10 300 GL column and snap-frozen in liquid nitrogen. The degree of labeling was determined at 280/496 nm.

## Biochemical assays

*Buffers.* All experiments were carried out in a common final assay buffer of the following composition if not stated otherwise: 20 mM Hepes pH 7.0, 100 mM KCl, 1.5 mM MgCl$_2$, 1 mM EGTA, 20 mM β-mercaptoethanol, 0.1 mM β-casein, 1 mM ATP. This buffer has a molar ionic strength of 0.133 M, which is close to the physiological ionic strength found in the literature (between 0.1 and 0.2 M).

*Functionalization of glass coverslip surfaces and protein immobilization.* For TIRF-M experiments, microscopy counter slides were passivated with PLL-PEG and coverslips (22 × 22 mm, 1.5 H, Marienfeld-Superior) functionalized as described[8,45]. Reaction flow chambers were blocked with a Pluronic block solution (0.1 mg/mL κ-casein, 1% Pluronic F-127, 1 mM TCEP, 1xKMEI) followed by 2 washing steps with 40 μl each (0.5 mM ATP, 1 mM TCEP, 1xKMEI, 0.1 mg/mL β-casein). Next, the channels were incubated with 75 nM streptavidin for 4 min following a washing step and incubation of 90 nM biotin-phalloidin[45,59] for 4 min.

Actin filament elongation assays using TIRF-M were performed as described[59,64]. Briefly, 9 μl of a 4,44x μM profilin-actin solution was incubated with 1 μl of 10x ME (0.5 mM MgCl$_2$, 2 mM EGTA) for 1 min at RT following the addition of 4 μl oxygen scavenging mix (1.25 mg/mL glucose-oxidase, 0.2 mg/mL catalase, 400 mM glucose)[65,66]. The Mg-ATP-profilin-actin was then mixed with 26 μl reaction buffer containing TIRF-buffer (20 mM Hepes pH 7.0, 100 mM KCl, 1.5 mM MgCl$_2$, 1 mM EGTA, 20 mM β-mercaptoethanol, 0.1 mg/mL β-casein, 0.2% methylcellulose (cP400, M0262, Sigma-Aldrich), 1 mM ATP, 2 mM Trolox) and additives including 10 nM Cy5-UTRN$_{261}$ and others, which are described in the specific methods and results sections. All TIRF-single filament elongation experiments were performed using profilin-actin as polymerizable substrate unless otherwise indicated in the corresponding results sections and figure legends.

## Measuring capping protein association and dissociation kinetics for the filament barbed end. Experiments were performed as described in the previous section with the following modifications:

$k_{on}$ *measurements of capping protein for filament barbed ends by TIRF-microscopy.* To determine the association rate constant for capping protein (wt and affinity mutants) for the actin filament barbed end, experiments were conducted similarly to protocols outlined in[67,68] with modifications. Briefly, small pre-polymerized phalloidin-stabilized actin seeds were immobilized to the glass surface of a TIRF chamber. Increasing capping protein (TMR-labeled or non-labeled CP) concentrations (0–6 nM, wt or mutant as indicated) were combined with 2 μM Mg-ATP actin monomers and 2 μM profilin1 in TIRF reaction buffer and flowed into the chamber. Elapsed time between mixing and imaging was ~40 s. The time courses of capping protein binding to the filament barbed ends (termination of barbed end polymerization) were recorded in single molecule or single filament TIRF time-lapse experiments. Only filaments that were visualized in the first frame were followed throughout the experiment. Filaments that were immobilized from solution during the experiment or new filaments that originated from spontaneous nucleation were excluded from the analysis. Therefore, only filament barbed ends that were initially free were scored. The experiment was recorded over a time period of 5 min in 1 s intervals. ≥80 filaments were measured for each capping protein concentration from ≥3 experiments.

*$k_{off}$ measurements of capping protein from filament barbed ends by TIRF-microscopy.* To determine the dissociation rate constant for capping protein (wt and affinity mutants) from the actin filament barbed end, experiments were conducted as described[67], with modifications. Briefly, pre-polymerized actin filaments were capped by mixing them with excess of capping protein (200 nM non-labeled CP, wt or mutant as indicated). After immobilizing the capped actin filaments in a flow chamber, capping protein was washed out by flushing the chamber with $2 \times 40\,\mu l$ of wash buffer. Following wash, $2\,\mu M$ of profilin-actin and 20 nM myotrophin/V1 (in TIRF buffer) were flushed into the chamber and the time courses of filament re-growth (capping protein dissociation) recorded in single filament TIRF time-lapse experiments. Elapsed time between mixing and imaging was ~40 s. Only filaments that were visualized in the first frame were followed throughout the experiment. Filaments that were immobilized from solution during the experiment or new filaments that originated from spontaneous nucleation were excluded from the analysis. Therefore, only filament barbed ends that were initially capped were scored in the analysis. The experiment was recorded over a time course of 50 min in 30 s intervals. ≥50 filaments were measured for each capping protein version (wt or mutant) from ≥3 experiments.

**Actin dendritic network assembly assay.** Actin dendritic network assembly assays were carried out using similar protocols as[5,44], with modifications. In brief, Streptavidin-coated polystyrene beads (Bangs laboratories, Ø =3 μm) were washed 2 times with NPF dilution buffer (20 mM Hepes pH 7.0, 100 mM KCl, 1 mM EDTA, 1.5 mM $MgCl_2$, 0.2 mg/mL β-casein, 5 mM β-mercaptoethanol) at 4 °C and spun at 8000 $g$ for 1 min to remove bead storage solution. After 5 min of sonication, the beads were incubated with biotinylated-NPF (mcherry-WAVE-PWCA and dark cherry-WAVE-PWCA, 1:40) for 30 min at 23 °C followed by 2 washing steps to remove non-specifically absorbed protein. To reduce non-specific protein absorption, glass counter surfaces and glass coverslips (22 × 22 mm, 1.5 H, Marienfeld-Superior) were plasma-treated and passivated with PLL-PEG. To generate actin dendritic network growth on NPF-coated beads, the following components were combined unless described otherwise (standard motility mix): 5 μM profilin-actin complexes (isolated via size-exclusion chromatography as described in the previous sections), 100 nM CP (20% TMR-labeled), 100 nM Arp2/3 (20% Alexa647-labeled), 15 nM Alexa488-lifeact, mCherry-NPF-coated beads and reaction buffer (20 mM Hepes pH 7.0, 100 mM KCl, 1.5 mM $MgCl_2$, 1 mM EGTA, 20 mM β-mercaptoethanol, 0.1 mg/mL β-casein, 0.2% methylcellulose (cP400, M0262, Sigma-Aldrich), 1 mM ATP). Based on the known affinity between profilin and cytoplasmic, mammalian actin ($K_D = 19$ nM,[45]), we can estimate that the total concentration of profilin-actin will be 4710 nM, whereas free profilin and actin will be about 290 nM at equilibrium under these conditions. To reduce the free actin concentration in control experiments (Supplementary Fig. 5), we added additional 0.5 μM profilin or 4.5 μM β-thymosin to the motility reaction. After 4 min (as indicated in the specific sections) at 23 °C, network growth was arrested by adding 15 μM latrunculinB and phalloidin to the mixture. Microscopy chambers were prepared by adding 5 μl of arrested bead motility mix between the glass items. The chambers were sealed with valap to prevent evaporation. After the beads had settled down on the glass surface, images were acquired in multicolor.

**FRET-WH2-competition assay to determine the NPF-WH2-occupancy on CP-decorated actin filament ends.** Assays were carried out similarly to the previous section with the following exceptions: Actin dendritic networks were grown from donor-labeled Alexa488-NPF (WAVE(PWCA)), CP and Arp2/3 were unlabeled, and 15 nM Cy5-UTRN261 was used as actin filament binding probe. The dendritic actin networks were grown for 5 min at 23 °C following an arrest cocktail containing 5 μM myotrophin/V1, 15 μM Latrunculin B, 15 μM phalloidin, 7.5 μM profilin1 and 7.5 μM quencher-labeled and Latrunculin B-stabilized A540Q-actin monomers (all final concentrations). The actin dendritic networks were directly imaged after adding the arrest cocktail. As a negative control, non-labeled wt profilin-actin instead of quencher-labeled actin was added to the arrest cocktail and directly mixed with 488-NPF-coated beads. As a positive control, only quencher-labeled profilin-actin was added to the 488-NPF-coated beads, assuming that all WH2 domains are free to interact only with quencher-labeled actin monomers in the absence of a dendritic network.

**Generation of short, uni-sized actin filaments decorated with capping protein.** To generate short (70–160 nm) actin filaments, actin monomers were freshly gel-filtered over a Superdex 200. 40 μl of 40–45 μM actin monomers were sufficient to start the polymerization reaction by addition of 25 μM capping protein. After 2 min incubation at 4 °C, 0.5 mM $MgCl_2$ and 0.2 mM EDTA final concentration were added for 1 min at 4 °C, followed by initiation of polymerization with 1xKMEI (10 mM imidazole pH 7.0, 50 mM KCl, 1.5 mM $MgCl_2$, 1 mM EGTA) for 2 min at 23 °C. To terminate the reaction, 48 μM phalloidin were added to the reaction mix followed by gelfiltration over a Superdex 200 increase 5/150 GL in 1xKMEI buffer. Short, unsized actin filaments could be collected from the first peak after 1 mL and were directly transferred onto cryo carbon grids for negative stain screening and cryo EM data collection. The freshly gelfiltered samples were stable for at least 4 hrs.

*Integration of protein intensity on SDS gel after SEC.* After SDS-gel run, actin and capping coomassie signal intensities were analyzed using ImageJ. After selecting protein signal areas for each protein, intensities were plotted in a plot profile reflecting the intensity signal across the selected area.

**Cell culture of B16-F1 cells.** B16-F1 cells and derived CapZa1/2 CRISPR#1, and CapZb KO#10 were cultured in DMEM (4.5 g/L glucose; Invitrogen, Germany), supplemented with 10% FCS (Gibco), 2 mM glutamine (Thermo Fisher Scientific) and penicillin (50 Units/mL)/streptomycin (50 μg/mL) (Thermo Fisher Scientific) at 37 °C and 7.5% $CO_2$. Cells were routinely transfected in 35 mm dishes or 6-well plates, using 500 ng DNA in total and 1 μL JetPrime. After overnight transfection, cells were either plated onto acid-washed, laminin-coated (25 μg/mL) coverslips for phalloidin or ArpC5A stainings or in μ-Slide 4 Well Ph + (Ibidi) chamber slides, coated with 25 μg/mL laminin, for live cell fluorescence microscopy.

**Generation of CapZα1/2 and CapZb CRISPR/Cas9 clones.** Disruption of CPα or –β expression in B16-F1 cells was achieved by targeting *CapZb* or *CapZa1* and *CapZa2* simultaneously using CRISPR/Cas9-mediated genome editing, using targeting/guide sequences CCTCAGCGATCTGATCGACC (for *CapZb* exon 2), GAGTTTAATGAAGTATTCAA (for *CapZa1* exon 2), and CAGAAGGAA-GATGGCGGATC (for *CapZa2* exon 1), cloned into plasmid pSpCas9(BB)-2A-Puro (Addgene. ID 48139). Plasmids were transfected into B16-F1 cells, followed by selection of transfected cells for three days using 2.5 μg/mL puromycin (Sigma-Aldrich, Taufkirchen, Germany). Puromycin-resistant cells were then extensively diluted and single cell-derived colonies grown for 5 to 7 days. After colony isolation, cells were expanded and clones screened by Western Blotting using CapZb or CapZa1 and CapZa2-specific antibodies, allowing selection of CRISPR-clones #10 (for *CapZb*) and #1 (for *CapZa1/2*). The genotypes of these clones were examined using TIDE sequencing[69]. In case of CapZb KO#10, all functional alleles could be disrupted. Instead, for CapZa1/2 CRISPR clone #1, sequencing revealed that all *CapZa1* alleles were disrupted, whereas expression from *CapZa2* alleles appeared partially possible from alleles harboring a 9 bp deletion. Additional, subsequent targeting of *CapZa2* alleles on exon 2 failed to eliminate expression from those 9 bp-deletion alleles.

**Western blotting.** For preparation of whole cell lysates, cells were washed with PBS, lysed using Laemmli-buffer (2% SDS, 10% glycerol, 5% β-mercaptoethanol, 0.05% bromophenol blue, 63 mM Tris-HCl pH 6,8), sonicated to shear genomic DNA and boiled for 5 min at 95 °C. Western blotting was carried out using standard techniques, and with antibodies as listed in the Key Resources Table (Supplementary Table 4). Chemiluminescence signals were obtained upon incubation with ECL Prime Western Blotting Detection Reagent (GE Healthcare), and recorded with ECL Chemocam imager (Intas, Goettingen, Germany). Original, uncropped versions of all blots used for figures in the paper can be found in Supplementary Fig. 9.

**Phalloidin and immune-stainings.** For sole stainings of the actin cytoskeleton of untransfected B16-F1, CapZa1/2 CRISPR#1 and CapZb KO#10 cells, respective cells were fixed with pre-warmed 4% paraformaldehyde in PBS, supplemented with 0.25 % glutaraldehye for 20 min, and permeabilized with 0.05% Triton X-100 in PBS for 30 s. The actin cytoskeleton was subsequently stained using Alexa 488-conjugated phalloidin (dilution 1:200 in PBS). For stainings upon rescue experiments of CapZb KO#10 cells with EGFP-tagged CapZb2 variants, B16-F1 wildtype and rescued CapZb2 KO#10 cells were fixed as described above with the only exception that glutaraldehyde was omitted from the fixation mixture. Subsequently, cells were either stained with ATTO-594-conjugated phalloidin (dilution 1:200 in PBS) or, after blocking with 5% horse serum / 1% BSA in PBS with anti-ArpC5A antibody (undiluted hybridoma supernatant; see Key Resources Table (Supplementary Table 4)). The primary antibody was then visualized with Alexa-594-conjugated anti-mouse IgG (1:100 dilution, see Key Resources Table (Supplementary Table 4)). Samples were mounted using ProLong Gold or VectaShield Vibrance antifade reagents and imaged using a ×63/1.4NA Plan apochromatic oil objective.

**Fluorescence microscopy data acquisition**

*TIRF-Microscopy data acquisition.* All in vitro experiments were performed at 23 ˚C, unless otherwise specified. Images were acquired using a customized Nikon Ti2 TIRF-Microscope with Nikon perfect focus system. TIRF-images were acquired by a dual EM CCD Andor iXon camera system (Cairn) controlled by NIS-Elements software. Dual color imaging was performed through an Apo TIRF 60x oil DIC N2 objective using a custom multilaser launch system (AcalBfi LC) at 488 nm and 560 nm and a Quad-Notch filter (400-410/488/561/631640). Images were acquired at intervals of 1–30 s.

*Wide-field epifluorescence-microscopy data acquisition.* All in vitro experiments were performed at 23°C. Bead motility assays were carried out using a Olympus IX81 wide field epifluorescence microscope with LED illumination (pE4000, CollLED). Image acquisition was performed by a Hamamatsu c9100-13 EMCCD

camera controlled by Micromanager 1.4 software[70]. Multi-color images were taken through an UPlanS APO 40x or 60x oil objective.

**Live cell fluorescence microscopy and quantification of protrusion velocity.** B16-F1, CapZa1/2 CRISPR#1 and CapZb KO#10 cells, untransfected or transfected with indicated EGFP-tagged CapZ constructs, were plated in μ-Slide 4 Well Ph + (Ibidi) chamber slides overnight. Thereafter, the medium was replaced with microscopy medium (F12 HAM HEPES-buffered medium, Sigma), including 10% FCS (Gibco), 2 mM glutamine (Thermo Fisher Scientific) and penicillin (50 Units/mL)/streptomycin (50 μg/mL) (Thermo Fisher Scientific). The slide was mounted onto an inverted microscope (Axiovert 100TV, Zeiss) equipped with a 37 °C incubator, and with an HXP 120 lamp for epifluorescence illumination, a halogen lamp for phase-contrast imaging, a Coolsnap-HQ2 camera (Photometrics) and electronic shutters driven by MetaMorph software (Molecular Devices). Unipolar cells were chosen for analysis. Phase contrast images, and fluorescence images in case of cells transfected with EGFP-tagged CapZ constructs, were acquired every 5 seconds using a 63x/1.4NA Plan apochromatic oil objective for a period of 5 min. Protrusion velocities were determined based on phase contrast-derived kymographs using MetaMorph 7.8. For every cell analyzed, kymographs at three different positions of the leading edge were taken. For EGFP fluorescence intensity measurements of transfected CapZb or CapZa1/2 cells shown in Supplementary Figure 8, average fluorescence intensities of large cytosolic regions were measured, and corrected for background by subtracting average fluorescence intensities of respective, adjacent extracellular regions. Obtained data were displayed as counts/pixel.

**Electron microscopy**

*Negative-stain microscopy.* Immediately after SEC, the quality of the short actin filaments preparation was assessed with negative stain EM. Briefly, 4 μL of sample were deposited onto a glow-discharged-carbon-coated copper grid and incubated for 30 s. After blotting the excess sample, the grid was washed twice with KMEI buffer and stained with 0.75% (w/v) uranyl formate. The grids were subsequently imaged at 120 kV in a Tecnai Spirit microscope equipped with a LaB$_6$ cathode and 4k × 4k CMOS detector F416 (TVIPS). Most importantly, we used these images to decide on the concentration used for preparing the Cryo-EM grids.

*Cryo grid preparation.* Cryo-EM samples were prepared using graphene-oxide-coated Quantifoil 1.2/1.3 300 mesh grids (GO-grids). We produce our GO-grids as follows: we took an aliquot of the 2 mg/mL stock graphene-oxide (GO) solution (Sigma-Aldrich) and sonicated it for 3 min in a bath sonicator. This step is necessary to separate aggregated GO plates. After this, we prepared dilutions between 0.05 and 0.15 mg/mL for screening. Quantifoil grids were freshly glow-discharged and 4 μL of our GO dilutions were immediately applied to the grids. After 3 min incubation, the grids were blotted from the back side using Whatman No. 5 and washed twice with 15 μL water. The quality of the resulting GO-grids was immediately assessed in our Tecnai Spirit microscope. Typically, we looked for a compromise between grid coverage and number of GO layers, with most selected grids showing a few layers—sometimes single ones—and almost 100% grid coverage. Shortly before applying the sample, the GO-grids were mildly glow discharged for 10 s using a 5 mA current.

Vitrification was performed in Vitrobot Mark IV. Four μL of sample were placed onto a GO-grid, incubated for 1 min at 13 °C and 100% humidity, blotted for 3.5 s with a force of –3, and subsequently plunged into liquid ethane.

*Data collection and preprocessing.* Cryo-EM images were obtained using a Talos Arctica microscope equipped with a 200 kV XFEG, and automatically collected using EPU. Exposures of 3 s were recorded with a Falcon III detector operated in linear mode, as stacks of 40 frames with a total dose of 60 e$^-$/Å$^2$. Due to the beam diameter of ~1.7 μm (50 μm C2 aperture) we collected a single exposure per hole. No objective aperture was used during data collection.

We collected a total of 4204 images in two separated sessions, covering a defocus range between −0.7 and −4.0 μm. Motion correction[71] and CTF estimation[72] were performed on the fly with TranSPHIRE[73].

*Cryo-EM image processing.* Most image processing steps were performed using SPHIRE[74]. Filament ends were automatically detected with crYOLO[75]. In order to improve the picking, we denoised the images using the noise2noise implementation within cryolo (janni) and further filtered the images using the ctfcorr.simple filter within the e2proc2d.py program of EMAN2[76] (See Supplementary Fig. 1). Using an ad-hoc model trained on these images, we obtained a total of 570,524 initial particles. In order to separate barbed from pointed ends, we performed two independent runs of 3D multi-reference alignment (sp_mref_ali3d.py) against references for both ends, and kept only those that we systematically assigned to the barbed end. The rest of the processing was performed in SPHIRE's filament mode. This was necessary since the high density of filament images provided too many local minima for alignment, which was alleviated by the rectangular mask used by helical SPHIRE. We used the final alignment parameters obtained in the multi-reference alignment step to estimate the in-plane rotation of the filaments. We then performed 2D classification with ISAC[77] and discarded all non-protein picks, but

kept all filaments, even if the class did not appear to be an end. This was necessary since, although ISAC produces validated and highly homogeneous classes, ends and full filaments could still be clearly seen mixed in some of them. With this set of clean barbed end particles, we started a 3D refinement using a full filament as a reference, followed by focused 3D classification. In cases where the ends were out of register, we re-centered the particles, re-extracted them, and re-run 3D refinement. After a few iterations, a single class containing capping protein could be obtained. We then exported the particles to RELION[78] where we performed Bayesian particle polishing[79], and re-imported the stack for a final round of k-means 2D clustering cleaning followed by local refinement within SPHIRE. Supplementary Fig. 1 shows a schematic summary of the procedure.

*Resolution estimation and maps postprocessing.* The final map was obtained from particles with some preferential orientation. To estimate the directional FSC of the map, we used the 3D-FSC package[80]. Local resolution was estimated using the false-discovery rate FSC method implemented in SPOC[81]. The final map was locally filtered using deepEMhancer[82]. Statistics of the Cryo-EM model of capped actin filaments are summarized in Supplementary Table 1.

**Atomic modeling.** We used MODELLER[83] to build a homology model of the complex between CP and the barbed end of human β-actin. As template, we used the capped end of the Arp1 filaments within the dynactin complex (PDBID:5ADX,[30]). We run this model through the Rosetta alanine scan serve[84] to identify side chains whose deletion would provide a similar decrease in binding affinity as the removal of the β tentacle.

To build an atomic model of the complex, we fitted the initial homology model into the density using a combination of ISOLDE[85] and Coot[86]. In order to deal with the strong quality gradient of our density, we used the locally filtered map produced by deepEMhancer for modeling. High-resolution refinement is possible in the area close to the center of the filament, where the resolution is highest. To deal with the least resolved areas, we used adaptive distance restraints during our ISOLDE runs. In this way, CP could be flexibly fitted into our density, but the high-resolution details were largely inherited from the initial model. Likewise, we manually included several distance restraints aimed at maintaining proper coordination of the Mg$^{2+}$ nucleotide complex at actin's active site. A final round of refinement in Phenix using positional restraints was used to remove some geometric violation and estimate the atomic B-factors, while keeping the overall structure produced by ISOLDE.

**Quantification and statistical data analysis.** All analyzed data were plotted and fitted in Origin9.0 G. All fluorescence microscopy experiments were analyzed in ImageJ manually via kymograph analysis and plotting of intensity profiles unless described otherwise.

*Determination of the association and dissociation rate constants of capping protein for actin filament ends.* Images were analyzed by manual filament tracking using the *segmented line tool* from ImageJ and further analyzed by the *kymograph plugin*. The slopes were measured to determine the polymerization rate of individual actin filaments. The pixel size/length was converted into microns/s. One actin monomer contributes to 2.7 nm of the actin filament length. For each experimental condition, the filament polymerization velocity was measured from ≥50 filaments per condition, and reported as mean value with error bars representing SD.

Time courses of the fraction of capped filament ends were fitted with mono-exponential functions to yield the observed pseudo-first order reaction rates ($k_{obs}$) as a function of capping protein concentration. The association rate constants ($k_{on}$) were determined from linear regression fits of the $k_{obs}$ mean values (error = SD) as a function of the total capping protein concentration [nM].

The time courses of the fraction of re-growing filament ends were fitted with single exponential functions to yield the dissociation rate constants ($k_{off}$).

The equilibrium dissociation constants ($K_D$) of capping protein for actin filament barbed ends were calculated from the measured dissociation rate constants ($k_{off}$) and association rate constants ($k_{on}$) using the following equation:

$$K_D = k_{off}/k_{on} \tag{1}$$

Errors for the equilibrium dissociation constants were calculated using error propagation.

*Quantification of fluorescence intensities in actin dendritic networks.* The mean fluorescence intensities of all network components (actin, CP and Arp2/3 complex) from multicolor fluorescence images were measured with ImageJ (ROI Manager->Multi Measure function) from line regions of interest (ROIs) matching the network area. The background fluorescence intensities for all components was determined from identical lines drawn adjacent to each network. The background signal was subtracted from the network signal. The integrated mean fluorescence intensities (representing polymerization, capping and nucleation rates) for each component were plotted as a function of distance from the bead surface [μm]. The rates for actin polymerization, capping and nucleation were calculated from integrated fluorescence intensities divided by reaction time.

*Quantification of the NPF-WH₂-occupancy on actin filament ends*. The fluorescence signal intensity of the Alexa488-labeled NPF was measured over the entire bead surface before and after addition of quencher-labeled Atto540Q actin monomers. The intensity data was plotted over the bead distance/diameter and represents the amount of $WH_2$ domains occupied by actin filament ends. For each condition, ≥25 dendritic actin networks from ≥3 experiments were quantified.

**Reporting summary**. Further information on research design is available in the Nature Research Reporting Summary linked to this article.

## Data availability

All quantitative data presented in the manuscript are contained in the source data file. Additional data are available from the corresponding authors upon request. A reporting summary for this Article is available as a Supplementary Information file. Our structure of capped barbed end is deposited in Protein Data Bank PDB, PDBID:7PDZ and Electron Microscopy Data Bank, EMDB-13343. The previously published structure of the dynactin complex[30] is available in the Protein Data Bank PDB, PDBID:5ADX. The previously published structure of monomeric actin in complex with the WH2 domain of WASP[51] is available in the Protein Data Bank PDB, PDBID:2A3Z. Source data are provided with this paper.

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

## Acknowledgements

We thank Philippe Bastiaens for continuous support, useful discussions and help in shaping the manuscript. We thank Natalie Petek and Dyche Mullins for Acanthamoeba castellani actin, as well as Dorothy Schafer for kindly providing EGFP-tagged CPβ2. We are also grateful to members of the Bieling lab and Thomas Surrey for comments on the manuscript. This work was supported by HSFP CDA00070/2017-2 (P.B.), Deutsche Forschungsgemeinschaft (DFG) Research Training Group GRK2223 (K.R.) and individual grant RO2414/8-1 (K.R.), the MaxSynBio network (P.B.), and the Max Planck Society (P.B. and S.R.).

## Author contributions

J.F. purified proteins, designed, performed and analyzed in vitro experiments under supervision of P.B. M.S. performed cell biology experiments under supervision of K.R. F.M. performed and analyzed all experiments related to electron microscopy under supervision of S.R. P.B. and S.R. conceived the project. All authors contributed to writing the manuscript.

## Funding

## Competing interests

The authors declare no competing interests.
