## [Peer Review File · Nature Communications]

REVIEWER COMMENTS

Reviewer #1 (Remarks to the Author):

In this study, the authors provide a new mechanism based on a structure of capped actin filaments to describe the effect of capping proteins on actin assembly. Overall, I think this is a very interesting study but I did find some inconsistencies in the kinetic analysis that the authors should address before publication.

In addition, capping protein is a potent actin nucleator of actin filaments growing from their pointed ends. In the motility assay, the free concentration of profilin is very important to avoid a competition between Arp2/3 complex nucleation at the bead surface and capping protein nucleation in the bulk. The authors mentioned 5 μ M actin/profilin complex but maybe I missed it but I could not find the concentration of actin monomers and profilin in the assay.

Comment on the kinetic analysis Figure 3.

The figure 3D is a variation of k_{obs} versus CP concentration. The slope of the plot is the k_+ , and the intercept with the Y axis the k_- . For the α Y277V β construct for example the k_- is around 0.01 S⁻¹ but in the table (based on Figure G) the k_- reported is 0.0026 about 4 times smaller. If now we report this value to calculate the K_d , we found about 1.3 nM K_d for the α Y277V β construct about 20 times lower affinity than wild type. This lower affinity will explain a lot the data from the bead motility assay. So, the authors should at least discuss this inconsistency.

In addition, the curve for WT in Figure 3D cannot have a negative intercept with the Y axis. The k_- should be positive.

Maybe the author should use the k_- determined in Figure 3G to impose the linear fit in Figure 3D.

Minor comment: In the materials and methods (maybe in other places) page 49, k_{off} should be with a small k not K .

Reviewer #2 (Remarks to the Author):

Capping protein (CP), also known as CapZ, is the quintessential actin filament barbed end capping protein in all cells, and because of this role it is a key player in nearly every form of actin network assembly and any function of the actin cytoskeleton. This is a beautiful work on the structure-function of capping protein (CP) that combines structural studies using cryo-EM, in vitro studies using TIRF microscopy and a reconstituted branched actin network assembly assay, and cellular studies using CRISPR/Cas9-mediated genome editing of CP subunits and re-expression of CP mutants. The major finding is that CP does more than just cap barbed ends. According to this study, CP also potentiates Arp2/3 complex-mediated actin assembly by an indirect mechanism, namely through competition with Nucleation-Promoting Factors (NPFs, which activate Arp2/3 complex) for binding to the barbed end of actin filaments. In other words, when CP is displaced from barbed ends (by proteins such as twinfilin, V-1 and CARMIL), NPFs can bind to the last actin subunit at the barbed end of the filament through their WH2 domain(s), and thus become unavailable for Arp2/3 complex nucleation. When CP is bound at the barbed end, its so-called beta tentacle masks the binding site for NPFs, and thus NPFs become available for Arp2/3 complex activation. This is an interesting and novel concept, and the paper is experimentally sound and rather complete. Therefore, I enthusiastically support publication of this work in Nature Communications. I do have some relatively minor comments, as well as a couple of questions that could be addressed through experiments (although not essential) and a proposal for rebalancing the message in a different way.

1. First, let's address the emphasis. When I was asked by NC to review this paper and read the Title and Abstract, I thought it was about the structure of CP at the barbed end, and while this would be meritorious enough, I felt it would not be particularly groundbreaking given what we already know from the structure of the dynactin complex in which CP is very well resolved. When I read the actual paper, I became increasingly excited by the results and the new model of CP function (mentioned above), and felt this paper could be far more impactful if these results and conclusions, i.e. the reconstituted branched actin network assembly and CP knockout experiments and conclusions, featured more prominently in the Title, Abstract and Introduction. Indeed, it is odd to claim that this novel mechanism of CP is structure-derived, because we have had the structure of CP at the barbed end of the Arp1 filament since 2015 and nobody came up with this idea before.

2. Concerning the structure itself, it is laudable and interesting the way this work solved the problem of resolving the barbed end of filaments with CP bound. This is the first time CP is resolved at relatively high resolution at the end of the actin filament. The prior study of Urnavicius et al, had it at the barbed end of the Arp1/beta-actin minifilament of dynactin. However, resolving the ends of the filament is challenging and, as a result, CP is poorly resolved in the current study (fragmented density) and much lower resolution (6-7Å) than the actin core. In this regard, I found the Atomic Modeling section in the Methods to have insufficient information. I can't imagine CP was all-atom refined based on this density. A better model would be obtained by simply rigid-body fitting the higher resolution structure obtained from the dynactin complex. Furthermore, the Carter lab published a higher resolution structure of dynactin than the one used here as reference, in which CP is beautifully resolved at ~3.5Å resolution (PDB code 6F1T; Urnavicius et al., Nature 2018). This should be the starting model, and probably fit simply as a rigid body (particularly that actin and Arp1

share ~51% sequence identity). Nevertheless, I would like to re-emphasize that what was accomplished here with the structure is difficult and commendable and confirms that CP binds actin and Arp1 filament barbed ends in the same manner.

3. Concerning the main message, i.e. that NPFs (WH2 domain) and CP (beta tentacle) compete for a common binding site on the last actin subunit at the barbed end, I have two questions that could possibly be addressed experimentally. First, what about the alpha tentacle? A deletion of the alpha-tentacle may have the same effect as the deletion of the beta-tentacle in actin network assembly by freeing a binding site for NPFs on the penultimate actin subunit at the barbed end. Why wasn't this tested? Presumably, a deletion of the alpha-tentacle has a more dramatic effect on the binding of CP, but this effect can be separated from network assembly, right? The kinetics-mimicking alpha tentacle mutants do not work the same way, because the alpha tentacle is still present and it may sterically hinder the binding of NPFs. So, to assume these alpha tentacle mutants have the same phenotype as WT CP is possible a misinterpretation of the data. Secondly, what about controlling for this effect in vitro through WH2 domain mutations? I understand that you need the WH2 domain for activation of Arp2/3 complex, and thus it cannot be easily mutated. But there could be a way around this; a debilitating mutation in the helix of the WH2 domain that would weaken its affinity for barbed ends. After all, the WH2 domain is a monomer-binding domain, and when actin becomes flatter in the filament its affinity is lower. This mutant would still bind monomers and assembly networks to a control level but may be unable to bind the barbed end. These are questions and controls to consider, although I don't think this is strictly necessary for acceptance of the paper.

Minor:

1. I suggest citing Yamashita et al., EMBO J 2003. This paper not only reported the structure of CP, but also developed the idea of the alpha-/beta-tentacles.
2. "Arp2/3" and "Arp2/3 complex" are both used. Please use one, preferentially Arp2/3 complex, which is the accepted name of the complex.
3. "terminal filament subunits" is mentioned often, and it appears to refer to subunit at the barbed end. However, because of the ambiguity with terminal subunits at the pointed end, this must be clarified, at least once.
4. P.4 "caps an short Arp1 filament" caps the short? Or caps a short?
5. The section "Deletion of the CP β tentacle modestly affects barbed end capping kinetics" is somewhat in disagreement with previous studies, specifically Wear et al. Curr Biol 2003. Could this disagreement be addressed head-on? Consider that in these TIRF assays the ratio of CP to barbed end is high, and thus CP would be expected to be mostly bound to barbed ends. In cells, however, the amount of CP could be limiting, and reducing its affinity 300-fold as reported by Wear et al (vs 3-fold reported here) could dramatically affect capping (independent of other effects on NPFs). Consider also that differences between TIRF and bulk assays typically go in the sense of bulk being correct. TIRF is based on event observation under conditions far removed from reality, and observations = event picking (in bulk all the events are treated as equal). Not to say TIRF is not a powerful tool, but possibly should not be used to estimate binding affinities.

6. P.8 “This is in part the result of its active removal from filaments ends via twinfilin”. Twinfilin is only one of many proteins that can do this (V-1, CARMIL and other CPI-containing proteins).

7. One thing to possibly mention is that twinfilin takes the place of the alpha and beta tentacles of CP at the barbed end (similar to what is proposed for WH2). So a beta-tentacle deletion should favor twinfilin binding, and the parallel is interesting.

Roberto Dominguez, PhD

University of Pennsylvania

Reviewer #3 (Remarks to the Author):

This interesting manuscript focuses on the interplay between Capping protein, Arp2/3, and its nucleation promoting factors (NPFs). By applying innovative cryoEM approaches, the authors determined the structure of capped actin filament barbed end. Importantly, by using elegant biochemical approaches, combined with mutagenesis and cell biology, they revealed that the WH2 domain –like beta-tentacle of Capping protein is critical for sustained nucleation of actin filaments. They also provide evidence that this is due masking the WH2 domain binding site of capped barbed end to prevent sequestration of the WH2 domain of NPFs to filament barbed ends.

This study is innovative, of good technical quality, and it provides important new insights into the function of Capping protein in actin dynamics. There are, however, few minor points that should be addressed to further strengthen this interesting manuscript.

1. The authors state in the ‘Discussion’ (page 14) that “the majority of soluble actin is bound to profilin in living cells (Funk et al., 2019; Kaiser et al., 1999), which itself competes with and reduces the monomer occupancy of NPF WH2 domains (Bieling et al., 2018).” Because some assays in the present study were performed by using profilin-actin (e.g. Fig. 4), the authors should more extensively discuss the possible effects of profilin/WH2 domain competition on the interpretation of their results.

2. The knockout-rescue experiments presented in Fig. 6 are interesting. However, the authors should also present some information about the expression levels of rescue constructs (i.e. were the wild-

type and mutant proteins expressed at comparable levels). Moreover, instead of just showing the localizations of wild-type and mutant Capping proteins in the rescue cells, it would be informative to show also the localization on Arp2/3 and density of F-actin at the leading edges of the rescue cells. Finally, the authors should discuss why the beta tentacle –truncated Capping protein no longer localizes to the leading edge (i.e. is this due to diminished filament branching at the leading edge of these rescue cells).

3. The manuscript would also benefit from more extensive characterization of the obtained structure. The authors could present a thorough comparison of their structure vs. the related dynactin complex structure, compare the conformations of the two terminal actin subunits in their structure vs. the structures of ADP-actin subunits within an actin filament, as well as discuss why the resolution of Capping protein was relatively low in certain regions of the protein in their structure.

4. On page 3, the authors state that Capping protein restricts the growth of filaments to short length, which prevents filament buckling for efficient force generation. This does not seem to make sense, because filaments capped by Capping protein no longer polymerize at their barbed ends, and thus cannot generate pushing force against the membrane. To avoid confusion, this sentence should be rewritten or deleted.

PL

REVIEWER COMMENTS

Reviewer #1 (Remarks to the Author):

In this study, the authors provide a new mechanism based on a structure of capped actin filaments to describe the effect of capping proteins on actin assembly. Overall, I think this is a very interesting study but I did find some inconsistencies in the kinetic analysis that the authors should address before publication.

We appreciate the positive response and thank the reviewer for the constructive criticism that helped us to improve the manuscript.

In addition, capping protein is a potent actin nucleator of actin filaments growing from their pointed ends. In the motility assay, the free concentration of profilin is very important to avoid a competition between Arp2/3 complex nucleation at the bead surface and capping protein nucleation in the bulk. The authors mentioned 5 μ M actin/profilin complex but maybe I missed it but I could not find the concentration of actin monomers and profilin in the assay.

We agree that the biochemical conditions of the motility assay were not described in sufficient detail previously. The referee correctly points out that CP can nucleate filaments if the concentration of bare actin monomers is sufficiently high and that such excess nucleation in bulk solution would be a concern. For our motility assays, we start with 1:1 complexes of mammalian cytoplasmic actin (β,γ isoforms) and profilin prepared by size-exclusion chromatography (Funk et al 2019 PMID: 31647411). Isolation of stoichiometric complexes is possible because mammalian cytoplasmic actin binds much more tightly ($K_D=19$ nM) to profilin compared to muscle actin, which is commonly used in the field (α isoforms, $K_D=100-1000$ nM depending on ionic strength, see Pring et al 1992 PMID: 1737036, Perelroizen et al 1994 PMID: 8031780). Based on the total concentration of profilin-actin (5 μ M) and the measured affinity, we can estimate that the free actin concentration in our motility assays will be only 290nM. This is too low to stimulate significant filament nucleation via CP in solution. For muscle actin, we would require about 13 μ M profilin to reduce the free monomer concentration to comparably low levels (assuming a K_D of 500nM), which is very close to what other labs are using (e.g. Boujemaa-Paterski et al PMID: 28935896). We have confirmed in previous work (Bieling et al 2016 PMID: 26771487) that motility assays under our conditions do not nucleate detectable filament content in solution in the absence of NPFs over tens of minutes. The reaction conditions are sufficiently stable to reconstitute branched networks with nearly constant assembly kinetics for hours, provided that the solution reservoir is large compared to the NPF-coated surface area. We have added additional text to the Results and Materials and Methods section of the manuscript to explain the biochemical conditions of our assays more clearly.

In addition, this comment together with a similar remark from the third reviewer motivated us to repeat the motility assays under two distinct conditions chosen to minimize the amount of free actin monomers in solution further (see Supplemental Figure 6). To this end, we added excess amounts of either profilin (500nM) or Thymosin- β_4 (4500nM) to 5 μ M profilin-actin. According to mass action, the free actin monomer concentration in solution should drop by about 50% (to 140nM) in response to these additions. Importantly, we find that the selective defects in branched network assembly resulting from the CP beta tentacle deletion are preserved in either condition. We conclude that these defects are robust towards changes in the actin monomer pool and are not due to a differential capacity of

these mutants to nucleate filaments in solution. We have included these findings as a new Supplemental Figure 6 and added corresponding text to the Results section of the paper.

Comment on the kinetic analysis Figure 3.

The figure 3D is a variation of k_{obs} versus CP concentration. The slope of the plot is the k_+ , and the intercept with the Y axis the k_- . For the alphaY277Vbeta construct for example the k_- is around 0.01 S⁻¹ but in the table (based on Figure G) the k_- reported is 0.0026 about 4 times smaller. If now we report this value to calculate the K_d , we found about 1.3 nM K_d for the alphaY277Vbeta construct about 20 times lower affinity than wild type. This lower affinity will explain a lot of the data from the bead motility assay. So, the authors should at least discuss this inconsistency. In addition, the curve for WT in Figure 3D cannot have a negative intercept with the Y axis. The k_- should be positive. Maybe the author should use the k_- determined in Figure 3G to impose the linear fit in Figure 3D.

We agree with the reviewer that the off-rates of CP from the filament barbed end should in principle also be obtainable from the y-intercepts of the k_{obs} data shown in Figure 3G. In reality, calculating dissociation rates in such a manner is notoriously error-prone, especially when k_{off} values are small and therefore close to the origin. Small errors in the data can easily lead to several-fold changes in k_{off} and even yield negative values, which should not be observed as correctly pointed out by the reviewer for some of our CP mutants. This is the reason why dissociation rates are most often measured in independent chase or buffer flow-out experiments as also done in our work. In our case, we trust the independently measured rate from the dissociation experiment much more, because it experimentally uncouples the processes of binding and dissociation of CP from the barbed end. To improve our kinetic analysis, we have followed the advice of the reviewer and included the independently measured off-rates in the k_{obs} data and fixed these values for the linear fits.

Minor comment: In the materials and methods (maybe in other places) page 49, k_{off} should be with a small k not K .

This error has been corrected.

Reviewer #2 (Remarks to the Author):

Capping protein (CP), also known as CapZ, is the quintessential actin filament barbed end capping protein in all cells, and because of this role it is a key player in nearly every form of actin network assembly and any function of the actin cytoskeleton. This is a beautiful work on the structure-function of capping protein (CP) that combines structural studies using cryo-EM, in vitro studies using TIRF microscopy and a reconstituted branched actin network assembly assay, and cellular studies using CRISPR/Cas9-mediated genome editing of CP subunits and re-expression of CP mutants. The major finding is that CP does more than just cap barbed ends. According to this study, CP also potentiates Arp2/3 complex-mediated actin assembly by an indirect mechanism, namely through competition with Nucleation-Promoting Factors (NPFs, which activate Arp2/3 complex) for binding to the barbed end of actin filaments. In other words, when CP is displaced from barbed ends (by proteins such as twinfilin, V-1 and CARMIL), NPFs can bind to the last actin subunit at the barbed end of the filament through their WH2 domain(s), and thus become unavailable for Arp2/3 complex nucleation. When CP is bound at the barbed end, its so-called beta tentacle masks the binding site for NPFs, and thus NPFs become available for Arp2/3 complex activation. This is an interesting and novel concept, and the paper is experimentally sound and rather complete. Therefore, I enthusiastically support publication of this work in Nature Communications. I do have some relatively minor comments, as well as a couple of questions that could be addressed through experiments (although not essential) and a proposal for rebalancing the message in a different way.

We thank the reviewer for his enthusiastic response and his thoughtful comments.

1. First, let's address the emphasis. When I was asked by NC to review this paper and read the Title and Abstract, I thought it was about the structure of CP at the barbed end, and while this would be meritorious enough, I felt it would not be particularly groundbreaking given what we already know from the structure of the dynactin complex in which CP is very well resolved. When I read the actual paper, I became increasingly excited by the results and the new model of CP function(mentioned above), and felt this paper could be far more impactful if these results and conclusions, i.e. the reconstituted branched actin network assembly and CP knockout experiments and conclusions, featured more prominently in the Title, Abstract and Introduction. Indeed, it is odd to claim that this novel mechanism of CP is structure-derived, because we have had the structure of CP at the barbed end of the Arp1 filament since 2015 and nobody came up with this idea before.

We agree with the reviewer that the configuration of CP at the actin filament barbed end we observe is very similar to its known structure within the dynactin complex (Urnavicius et al 2015 and 2018, PMIDs: 25814576,29420470) and to computational predictions (Kim et al. 2010, PMID: 20969875). In hindsight, the new mechanism of CP function we put forth might have been predicted from existing structural data alone. However, confirming this binding mode first and demonstrating that both CP tentacles are indeed docked to the hydrophobic clefts of actin at the barbed end was crucial. We would not have invested two years of work to obtain the structure if we had believed otherwise. Arp1 and actin are sufficiently divergent (~50% sequence identity) that the two proteins have clearly different properties (see for instance Bingham and Schroer 1999, PMID: 10074429). Most importantly, both protein diverge in regions involved in capping protein binding. As such, an undocked state of the tentacle -while admittedly unlikely- might have been possible. For clarity – and as also requested by

reviewer 3 – we have added an additional supplementary figure (Supplemental Figure 3) with the comparison between dynactin and our complex.

In addition, we made further changes to the Title, Abstract and Introduction sections to shift the focus from the structure to the biochemical and cell biochemical aspects and their functional implications as requested.

2. Concerning the structure itself, it is laudable and interesting the way this work solved the problem of resolving the barbed end of filaments with CP bound. This is the first time CP is resolved at relatively high resolution at the end of the actin filament. The prior study of Urnavicius et al, had it at the barbed end of the Arp1/beta-actin minifilament of dynactin. However, resolving the ends of the filament is challenging and, as a result, CP is poorly resolved in the current study (fragmented density) and much lower resolution (6-7Å) than the actin core. In this regard, I found the Atomic Modeling section in the Methods to have insufficient information. I can't imagine CP was all-atom refined based on this density. A better model would be obtained by simply rigid-body fitting the higher resolution structure obtained from the dynactin complex. Furthermore, the Carter lab published a higher resolution structure of dynactin than the one used here as reference, in which CP is beautifully resolved at ~3.5Å resolution (PDB code 6F1T; Urnavicius et al., Nature 2018). This should be the starting model, and probably fit simply as a rigid body (particularly that actin and Arp1 share ~51% sequence identity). Nevertheless, I would like to re-emphasize that what was accomplished here with the structure is difficult and commendable and confirms that CP binds actin and Arp1 filament barbed ends in the same manner.

We agree that the level of detail presented in the atomic modeling section of the methods section was previously insufficient. We have improved this in the current version of the manuscript. As correctly pointed out by the reviewer, the resolution and quality of the map for CP is not high enough for unrestrained full atom refinement. What we have done instead is to use adaptive distance restraints within ISOLDE to fit CP into the density without losing the high-resolution details from the source model. We used a similar strategy to model the MgADP-Pi ligands, where we placed distance restraints to keep actin's active site from collapsing. Later refinement within Phenix used also positional restraints, and was only meant to locally fix geometrical problems with the model coming from the MDFF refinement. We believe that a rigid body fit of the CP model from 6F1T would not improve significantly our structure, and would certainly introduce clashes that would need to be additionally refined. In addition, while pig (from dynactin) and mouse (from our work) CP are almost identical, there are a few amino acid differences in CapA, requiring anyways some sort of modeling.

3. Concerning the main message, i.e. that NPFs (WH2 domain) and CP (beta tentacle) compete for a common binding site on the last actin subunit at the barbed end, I have two questions that could possibly be addressed experimentally. First, what about the alpha tentacle? A deletion of the alpha-tentacle may have the same effect as the deletion of the beta-tentacle in actin network assembly by freeing a binding site for NPFs on the penultimate actin subunit at the barbed end. Why wasn't this tested? Presumably, a deletion of the alpha-tentacle has a more dramatic effect on the binding of CP, but this effect can be separated from network assembly, right? The kinetics-mimicking alpha tentacle mutants do not work the same way, because the alpha tentacle is still present and it may sterically hinder the binding of NPFs. So,

to assume these alpha tentacle mutants have the same phenotype as WT CP is possible a misinterpretation of the data.

This is an interesting suggestion, which we had not considered previously because of two reasons: i) Prior work indicates that a complete alpha tentacle deletion dramatically inhibits barbed end binding as pointed out by the reviewer (Wear et al 2003, PMID: 12956956). ii) The NPF WH2 docked to the penultimate actin subunit does not only overlap with the alpha tentacle, but comes in immediate proximity to the body of the CP beta subunit. Careful inspection of the structure, however, indicated that removing the very end of the alpha tentacle might potentially suffice to vacate this NPF binding site (see Supplemental Figure 8). To test this hypothesis, we created two additional CP variants that removed the last 8 or 9 AAs of the alpha tentacle. We tested the effect of these deletions in reconstituted branched network assembly (see Supplemental Figure 8). Both of those CP variants, however, behaved like wildtype capping protein, indicating that they were active in barbed end binding as anticipated, but did not inhibit nucleation like the beta tentacle deletion. We believe the latter finding reflects either of two possibilities: A) The short deletions do not free the WH2 site at the penultimate subunit entirely, potentially because of additional clashes with the CP body might exist as explained above. B) Ultimate and penultimate actin subunits are differentially conducive to NPF WH2 binding. While we do not observe obvious structural differences between these subunits, the penultimate subunit will be more distant (by about 2.5nm) from the surface-immobilized NPF on average. This alone might result in preferential binding of WH2 to the last subunit. We have added all these data as a new Supplemental Figure 8 and included them into the Results section of the paper.

Secondly, what about controlling for this effect in vitro through WH2 domain mutations? I understand that you need the WH2 domain for activation of Arp2/3 complex, and thus it cannot be easily mutated. But there could be a way around this; a debilitating mutation in the helix of the WH2 domain that would weaken its affinity for barbed ends. After all, the WH2 domain is a monomer-binding domain, and when actin becomes flatter in the filament its affinity is lower. This mutant would still bind monomers and assembly networks to a control level but may be unable to bind the barbed end. These are questions and controls to consider, although I don't think this is strictly necessary for acceptance of the paper.

This is an interesting point as well. We are aware that the Taunton and Way labs have used such variants for N-WASP previously to reduce the binding of WH2 domains to filament ends (Co et al 2007 PMID: 17350575, Weisswange et al 2009 PMID: 19262673). We generated the corresponding individual R506A or R512Q substitutions in the WH2 domain of WAVE1. However, when tested for Arp2/3 activation in bulk pyrene assays at conditions similar to our bead motility reconstitutions (5 μ M profilin-actin), we discovered that both of these WH2 variants were significantly less active compared to wildtype WAVE1 (see Referee Figure 1). Notably, Co et al arrived at different conclusions for N-WASP using bare actin monomers in the absence of profilin in branching assays. We believe that this difference can be explained by the inhibitory effect of profilin on actin binding to NPF WH2 (Bieling et al 2018, PMID: 29141912). Taken together, we concluded that these mutants are impaired in Arp2/3 activation under realistic biochemical conditions (high profilin and low free actin concentrations) and therefore cannot be expected to functionally uncouple barbed end binding from Arp2/3 activation as commonly assumed. Hence, we did not test for their effect in reconstituted bead motility in the presence of CP mutants.

Minor:

1. I suggest citing Yamashita et al., EMBO J 2003. This paper not only reported the structure of CP, but also developed the idea of the alpha-/beta-tentacles.

We apologize for this oversight. We have now cited the original Maeda paper as requested.

2. “Arp2/3” and “Arp2/3 complex” are both used. Please use one, preferentially Arp2/3 complex, which is the accepted name of the complex.

We now use “Arp2/3 complex” throughout the paper.

3. “terminal filament subunits” is mentioned often, and it appears to refer to subunit at the barbed end. However, because of the ambiguity with terminal subunits at the pointed end, this must be clarified, at least once.

We have clarified this in the Abstract and several other prominent parts of the manuscript.

4. P.4 “caps an short Arp1 filament” caps the short? Or caps a short?

We fixed this typo.

5. The section “Deletion of the CP β tentacle modestly affects barbed end capping kinetics” is somewhat in disagreement with previous studies, specifically Wear et al. Curr Biol 2003. Could this disagreement be addressed head-on? Consider that in these TIRF assays the ratio of CP to barbed end is high, and thus CP would be expected to be mostly bound to barbed ends. In cells, however, the amount of CP could be limiting, and reducing its affinity 300-fold as reported by Wear et al (vs 3-fold reported here) could dramatically affect capping (independent of other effects on NPFs). Consider also that differences between TIRF and bulk assays typically go in the sense of bulk being correct. TIRF is based on event observation under conditions far removed from reality, and observations = event picking (in bulk all the events are treated as equal). Not to say TIRF is not a powerful tool, but possibly should not be used to estimate binding affinities.

We do not believe that our TIRF-based measurements are in disagreement with earlier Cooper lab work or other previous data. Wear et al (PMID: 12956956) generated two differently long truncations (delta 34 and 28) of the beta tentacle. Our beta tentacle deletion (delta 23) is even slightly shorter and should thus be compared to their delta28 variant, for which they report a drop in barbed end affinity by about 20-30-fold depending on experimental method. This is quite similar to the 6-fold drop we observe. The residual 3-5-fold difference might be due to differences in truncation length, ionic strength of the buffer (50 vs 100 mM monovalent salts) or the use of muscle over cytoplasmic actin.

We do not fully agree with the reviewer concerning the precision of TIRFM single molecule vs bulk kinetic assays in this specific case, because also the bulk assays that can be used to determine rates for this particular interaction have strong limitations. These bulk measurements are quite indirect, because rates are calculated from a kinetic model of seeded barbed end growth, which in itself requires several simplifying assumptions (no pointed end growth, no filament nucleation via CP etc). To illustrate the degree of uncertainty in these particular bulk experiments, we have taken the part of the original data (scanned from figure images) for wildtype CP from Wear and fitted them with their kinetic model either using dissociation rates that differ by two orders of magnitude (0.0018 s^{-1} , 0.00018 s^{-1} or 0.000018 s^{-1} , Figure 2 for the referee). One can appreciate that the difference in fit quality is quite marginal despite the large difference in dissociation rate assumed. In conclusion, we believe that our observations are in line with previous data and that the direct assays we employed are sufficiently precise to kinetically characterize CP and its variants.

Referee Figure 2: Inhibition of seeded actin growth by CP according to Wear et al, globally fitted with fixed dissociation constants at indicated values.

6. P.8 “This is in part the result of its active removal from filaments ends via twinfilin”. Twinfilin is only one of many proteins that can do this (V-1, CARMIL and other CPI-containing proteins).

We have revised this sentence to clarify that active uncapping is a shared ability of several allosteric CP regulators and cited the Edwards et al review (PMID: 25207437).

7. One thing to possibly mention is that twinfilin takes the place of the alpha and beta tentacles of CP at the barbed end (similar to what is proposed for WH2). So a beta-tentacle deletion should favor twinfilin binding, and the parallel is interesting.

We have added a short paragraph to the Discussion section, which takes up this interesting point. Accelerated uncapping through twinfilin in the absence of the beta tentacle might contribute to the strongly reduced localization of this CP variant to lamellipodia.

Roberto Dominguez, PhD
University of Pennsylvania

Reviewer #3 (Remarks to the Author):

This interesting manuscript focuses on the interplay between Capping protein, Arp2/3, and its nucleation promoting factors (NPFs). By applying innovative cryoEM approaches, the authors determined the structure of capped actin filament barbed end. Importantly, by using elegant biochemical approaches, combined with mutagenesis and cell biology, they revealed that the WH2 domain –like beta-tentacle of Capping protein is critical for sustained nucleation of actin filaments. They also provide evidence that this is due masking the WH2 domain binding site of capped barbed end to prevent sequestration of the WH2 domain of NPFs to filament barbed ends.

This study is innovative, of good technical quality, and it provides important new insights into the function of Capping protein in actin dynamics. There are, however, few minor points that should be addressed to further strengthen this interesting manuscript.

We thank the reviewer for appreciating our work and his constructive feedback.

1. The authors state in the 'Discussion' (page 14) that “the majority of soluble actin is bound to profilin in living cells (Funk et al., 2019; Kaiser et al., 1999), which itself competes with and reduces the monomer occupancy of NPF WH2 domains (Bieling et al., 2018).” Because some assays in the present study were performed by using profilin-actin (e.g. Fig. 4), the authors should more extensively discuss the possible effects of profilin/WH2 domain competition on the interpretation of their results.

We do agree with the reviewer that competition between NPF WH2 and profilin might affect the rate of nucleation via the Arp2/3 complex depending on profilin levels, which might affect our results. Motivated by this and similar comments from the first referee, we have repeated our bead motility experiments under two distinct conditions chosen to decrease monomer loading of NPF WH2 (see Supplemental Figure 6). To this end, we added excess amounts of either profilin (500nM) or Thymosin- β_4 (4500nM) to 5 μ M profilin-actin. According to mass action, the free actin monomer concentration in solution should drop by about 50% (from 290 to 140nM) in response to these additions. Importantly, while the overall nucleation kinetics might change slightly, we find that the selective defects in branched network assembly resulting from the CP beta tentacle deletion are preserved. We therefore deduced that these defects are robust towards changes in the actin monomer pool, indicating that the mechanism we uncovered does not hinge on a specific level of free actin monomers as controlled by profilin. We have included these findings as new Supplemental Figure 6, and described them as well as discussed their implications in the Results and Discussion sections of the paper, respectively.

2. The knockout-rescue experiments presented in Fig. 6 are interesting. However, the authors should also present some information about the expression levels of rescue constructs (i.e. were the wild-type and mutant proteins expressed at comparable levels). Moreover, instead of just showing the localizations of wild-type and mutant Capping proteins in the rescue cells, it would be informative to show also the localization on Arp2/3 and density of F-actin at the leading edges of the rescue cells. Finally, the authors should discuss why the beta tentacle –truncated Capping protein no longer localizes to the leading edge (i.e. is this due to diminished filament branching at the leading edge of these rescue cells).

These are important points. We have now determined the global expression levels of our EGFP-tagged rescue constructs by Western Blotting, and also quantified fluorescence intensities at the single cell level from live cell widefield imaging data (see new Supplemental Figure 9A,B). These analyses confirmed that all of our CP mutants are expressed at levels that are very close to their corresponding wildtype rescue controls. Hence, observed differences in rescue efficiency between distinct constructs cannot be explained by their differential expression.

In addition, we have analyzed Arp2/3 complex and actin filament densities at the leading edges of rescue cells expressing either wildtype or beta tentacle-truncated CP, as requested (see new Supplemental Figure 9C). Note that for those analyses, we are focusing on flat, horizontally protruding lamellipodial structures as opposed to ruffling ones, to avoid confusion with accumulation of fluorescence at the cell periphery due to sole cytoplasmic thickening. Doing this, we have confirmed that the levels of both Arp2/3 complex and filamentous actin in lamellipodia are substantially reduced, albeit not abolished in the absence of the CP beta tentacle, as expected. These *in vivo* effects are in line with our model and mirror those in our reconstitutions, in which we see a similarly strong reduction of both components of branched networks. We therefore concluded that the virtual absence of beta tentacle-truncated CP from flat, protruding lamellipodia-like networks is at least in part due to strongly diminished, Arp2/3-dependent branching in those cells. Whether other unrelated effects -such as a potential accelerated uncapping of beta tentacle-truncated CP via proteins like twinfilin- potentially also contributes to its suppression of leading edge accumulation, certainly constitutes an attractive possibility worth exploring in future studies. We mention this possibility in the Discussion section of our paper.

3. The manuscript would also benefit from more extensive characterization of the obtained structure. The authors could present a thorough comparison of their structure vs. the related dynactin complex structure, compare the conformations of the two terminal actin subunits in their structure vs. the structures of ADP-actin subunits within an actin filament, as well as discuss why the resolution of Capping protein was relatively low in certain regions of the protein in their structure.

We have added a new supplementary figure (Supplementary Figure 3) in which we compare the structures of the dynactin complex and our capped filaments. We have also added a small paragraph in the results section discussing the comparatively lower resolution of capping protein in the map. The issue of possible differences between the terminal protomers and internal subunits is indeed very interesting. We have refrained from a detailed comparison for a number of reasons: (i) the terminal protomers in our structure have a different nucleotide state (see Fig S2) and no phalloidin at their interface, possibly confounding changes due to their position along the filament, (ii) there is a strong resolution gradient which could obscure potential small differences, which are typically for actin in different filamentous states (iii) it is only marginally related to the main topic of the paper which is the role of CP in stimulating Arp2/3-dependent nucleation. We are currently preparing a separate manuscript dealing with the in-depth comparison of the structure of actin at distinct filament ends.

4. On page 3, the authors state that Capping protein restricts the growth of filaments to short length, which prevents filament buckling for efficient force generation. This does not seem to make sense, because filaments capped by Capping protein no longer polymerize at their barbed ends, and thus cannot

generate pushing force against the membrane. To avoid confusion, this sentence should be rewritten or deleted.

We do not fully agree with the reviewer here, because several recent studies (Akamatsu et al 2020 PMID: 25207437, Jasnin et al 2021 preprint) indicate that capped filaments can exert elastic forces on membranes and thus contribute to force generation even when not actively polymerizing. In addition, by setting a limit to filament length, CP ensures that dominantly short filament contact the membrane. If filaments beyond a certain length would not be capped, they would continue to elongate while buckling, thus resulting in futile polymerization. Nonetheless, we have rephrased this sentence to convey the point we wanted to make more clearly.

PL

REVIEWERS' COMMENTS

Reviewer #1 (Remarks to the Author):

The authors' response and revisions have satisfactorily addressed my comments on the earlier version of the manuscript.

Reviewer #2 (Remarks to the Author):

The author's response is satisfactory. I look forward to seeing this work its final published form.

Reviewer #3 (Remarks to the Author):

The authors have satisfactorily addressed my previous comments. This manuscript presents important new information on the mechanisms of actin dynamics, and it is in my opinion now suitable for publication.

Pekka Lappalainen, Univ. Helsinki